# Micropattern differentiation of mouse pluripotent stem cells recapitulates embryo regionalized cell fate patterning

Sophie M Morgani[1,2], Jakob J Metzger[3], Jennifer Nichols[2], Eric D Siggia[3]*, Anna-Katerina Hadjantonakis[1]*

[1]Developmental Biology Program, Sloan Kettering Institute, Memorial Sloan Kettering Cancer Center, New York, United States; [2]Wellcome Trust-Medical Research Council Centre for Stem Cell Research, University of Cambridge, Cambridge, United Kingdom; [3]Center for Studies in Physics and Biology, The Rockefeller University, New York, United States

**Abstract** During gastrulation epiblast cells exit pluripotency as they specify and spatially arrange the three germ layers of the embryo. Similarly, human pluripotent stem cells (PSCs) undergo spatially organized fate specification on micropatterned surfaces. Since in vivo validation is not possible for the human, we developed a mouse PSC micropattern system and, with direct comparisons to mouse embryos, reveal the robust specification of distinct regional identities. BMP, WNT, ACTIVIN and FGF directed mouse epiblast-like cells to undergo an epithelial-to-mesenchymal transition and radially pattern posterior mesoderm fates. Conversely, WNT, ACTIVIN and FGF patterned anterior identities, including definitive endoderm. By contrast, epiblast stem cells, a developmentally advanced state, only specified anterior identities, but without patterning. The mouse micropattern system offers a robust scalable method to generate regionalized cell types present in vivo, resolve how signals promote distinct identities and generate patterns, and compare mechanisms operating in vivo and in vitro and across species.
DOI: https://doi.org/10.7554/eLife.32839.001

*For correspondence:
siggiae@mail.rockefeller.edu
(EDS);
hadj@mskcc.org (A-KH)

**Competing interests:** The authors declare that no competing interests exist.

## Introduction

Gastrulation is the process of coordinated cell fate specification, spatial patterning and morphogenesis that establishes the blueprint of the adult organism. During gastrulation, the pluripotent epiblast (Epi) differentiates into the three definitive germ layers of the embryo; the ectoderm, mesoderm and endoderm. In the mouse, these events are initiated at approximately embryonic day (E) 6.25 by a convergence of signals, emanating from both extraembryonic and embryonic tissues, acting at the proximal, posterior of the embryo. The resulting BMP/Wnt/Nodal/FGF signaling hub drives posterior Epi cells to undergo an epithelial-to-mesenchymal transition (EMT) (*Hashimoto and Nakatsuji, 1989*; *Tam and Loebel, 2007*; *Arnold and Robertson, 2009*), establishing a dynamic territory referred to as the primitive streak (PS). The PS elongates and extends distally as gastrulation proceeds. Distinct cell types are specified depending on the time and position at which they undergo EMT and exit the PS (*Kinder et al., 1999*; *Lawson, 1999*). Emerging mesenchymal cells either move proximally and laterally, forming the extraembryonic mesoderm, or bilaterally in an anterior direction circumnavigating the space between the Epi and outer visceral endoderm (VE) layers, giving rise to the embryonic mesoderm and definitive endoderm (DE). Epi cells that maintain an epithelial state and do not ingress through the PS form the ectoderm.

Pluripotent stem cells (PSCs) are the in vitro counterpart of the pluripotent Epi of the embryo. They can be expanded indefinitely and differentiated into derivatives of all germ layers

(*Keller, 2005*). Standard differentiation protocols generate cell fates in a spatially disorganized manner that is incomparable with in vivo gastrulation. However, it was recently it was shown that, when human embryonic stem cells (hESCs) were differentiated within geometrically uniform, circular micropatterns, they reproducibly patterned cell fates with radial symmetry (*Warmflash et al., 2014*; *Tewary et al., 2017*; *Etoc et al., 2016*). Based on a limited number of markers, hESC micropatterned colonies were suggested to give rise to a central ectoderm population followed by concentric circular territories of mesoderm, endoderm, and an outer trophectoderm layer (*Warmflash et al., 2014*). These findings revealed the capacity of the BMP, Wnt and Nodal signaling pathways to collectively organize cell fates. The scalability and reproducibility of this assay coupled with the ease of genetically modifying PSCs, the ability to manipulate culture conditions and the simplicity of imaging make this a robust and attractive system to disentangle the cellular behaviors and signaling interactions that pattern mammalian embryos. Even so, this human organotypic system raised many questions, largely due to the absence of a human in vivo standard for direct comparison and assignment of cell identities.

Here we adapted the micropattern-based system to defined medium conditions to precisely dissect signaling requirements, and to mouse PSCs for which in vivo reference points are accessible to assign cell fates. We first converted mouse ESCs to epiblast-like cells (EpiLCs), the in vitro counterpart of the Epi of the early pre-gastrulation embryo (*Hayashi et al., 2011*). Mouse EpiLCs seeded onto circular micropatterned surfaces formed a simple epithelial morphology in a flat-disc geometry. By all markers examined, these cells were identical to the Epi of the E5.5-E6.0 embryo. When exposed to gastrulation-promoting factors, micropatterned EpiLCs underwent an EMT and recapitulated organized germ layer differentiation of specific regions of the mouse gastrula. This demonstrated that the cup-shaped geometry of the rodent embryo is not requisite for the spatial patterning of mouse pluripotent cells. Furthermore, the capacity to undergo spatially organized germ layer differentiation under these conditions was specific to EpiLCs. Under the same conditions neither ESCs, corresponding the pre-implantation Epi, nor epiblast stem cells (EpiSCs), corresponding to the gastrulating Epi (*Figure 1A*), demonstrated robust cell fate patterning. Hence, the mouse micropattern system offers a defined and quantitative tool to functionally assess the spectrum of described mouse pluripotent states (*Morgani et al., 2017*).

In vivo, we observed a proximal-to-distal gradient of BMP signaling activity – cells in the posterior (proximal) PS exhibited high signaling activity, while those in the anterior (distal) PS showed no activity. We hypothesized that by modulating the signals provided to mouse PSCs we could recapitulate the proximal-distal environments operative in vivo and generate distinct regional identities in vitro. Exposure of micropatterned EpiLCs to posterior signals, BMP, FGF, ACTIVIN (NODAL) and WNT, promoted an EMT and acquisition of posterior Epi, PS, embryonic and extraembryonic mesoderm identities. When BMP was removed, emulating the anterior PS environment (in which FGF, ACTIVIN and WNT are acting), anterior Epi, anterior PS and/or AxM and DE cell types were formed.

Hence, we demonstrated for the first time that in vitro micropattern differentiation parallels events occurring during gastrulation in vivo in mammalian embryos, and that mouse PSCs residing in a flat-disc geometry can pattern cohorts of neighboring regional identities correlating with those established in the embryo. Utilizing the micropattern system to manipulate the BMP pathway in isolation allowed us to extend findings made in mouse mutants by addressing the anterior versus posterior requirements for this signaling pathway within the PS. We established a direct requirement for BMP4 in posterior mesoderm formation, and demonstrated that BMP signaling is not required for DE and anterior PS/AxM specification. Further quantitative analysis of the signaling dynamics, the role of secreted inhibitors and cell-cell interactions should reveal how pathways operate in a flat-disc-shaped geometry, resembling the majority of mammalian embryos (including human), that can now be directly correlated to mouse, the most developed mammalian genetic model.

## Results

### Micropatterned EpiLCs correspond to the pre-gastrulation epiblast

The pluripotent state is a continuum spanning from Epi cell specification in the pre-implantation blastocyst (at approximately E3.5) to differentiation at gastrulation which initiates at E6.25 (*Morgani et al., 2017*) (*Figure 1A*). Prior to the onset of gastrulation (E5.5-E6.0), the Epi is in a

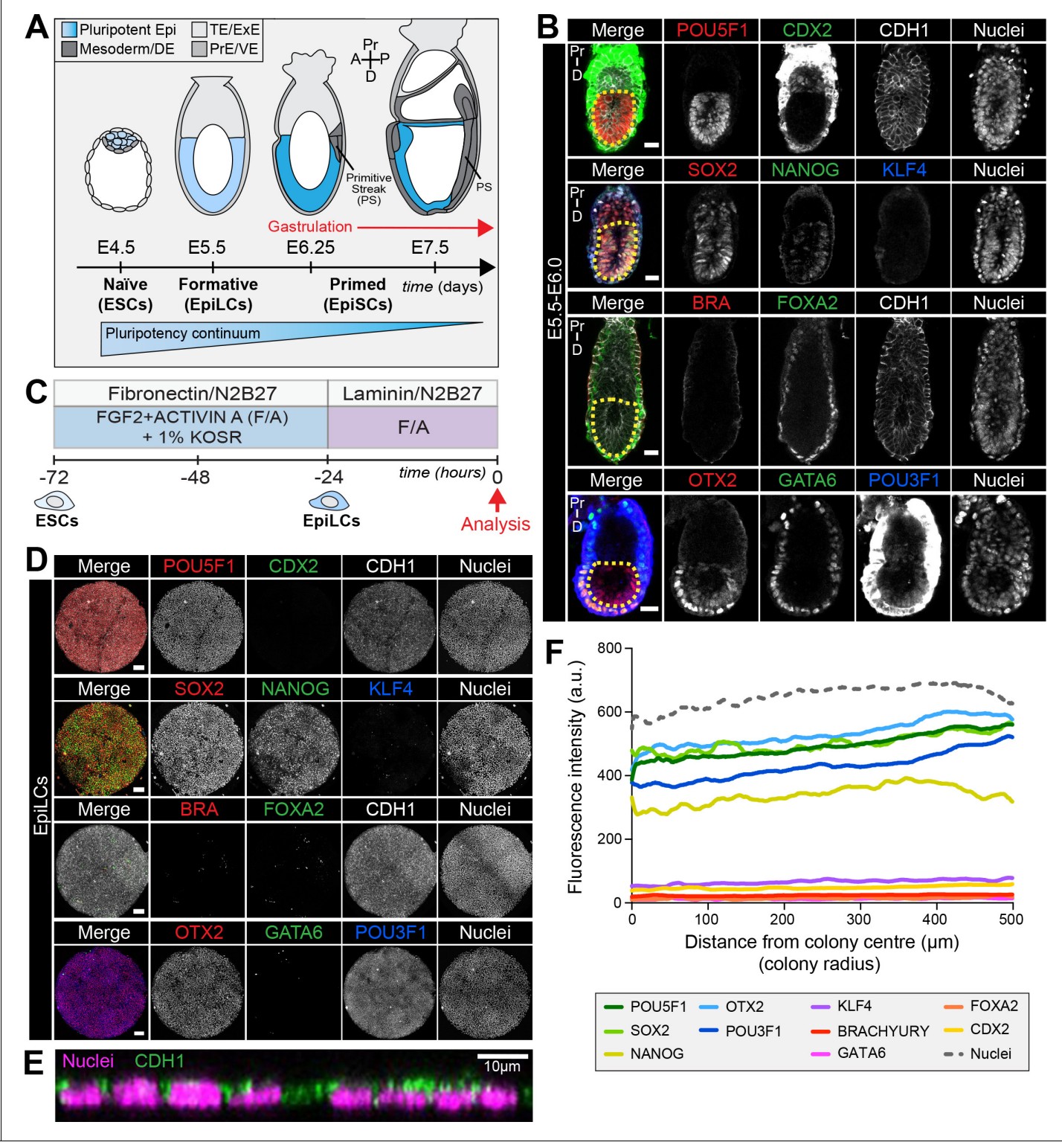

**Figure 1.** EpiLCs represent a pluripotent state correlating to the pre-streak epiblast of the embryo. (**A**) Development of the mouse pluripotent epiblast (Epi) from embryonic day (E) 4.5 to 7.5 and correlating in vitro pluripotent states. ESCs, embryonic stem cells; EpiLCs, epiblast-like cells; EpiSCs, epiblast stem cells; TE/ExE, trophectoderm/extraembryonic ectoderm; PrE/VE, primitive endoderm/visceral endoderm; DE, definitive endoderm; A, anterior; P, posterior; Pr, proximal; D, distal. (**B**) Sagittal sections of immunostained E5.5-E6.0 embryos. Yellow dashed line demarcates Epi. Scale bars, 25 μm. Non-nuclear anti-BRACHYURY/CDX2/POU3F1 VE fluorescence represents non-specific binding. (**C**) ESCs were converted to EpiLCs on Fibronectin in N2B27 with FGF2 and ACTIVIN A (F/A) and knockout serum replacement (KOSR) for 48 hr. EpiLCs were plated onto Laminin-coated

*Figure 1 continued on next page*

*Figure 1 continued*

micropatterns overnight and analyzed the following day (0 hr). (**D**) Maximum intensity projections of immunostained 1000 µm diameter EpiLC micropatterned colonies. Scale bars, 100 µm. (**E**) Confocal image showing a z-axis (side profile) region of an immunostained EpiLC micropatterned colony. (**F**) Quantification of immunostaining voxel fluorescence intensity from center (0) to edge (500). Data represents average voxel intensity across multiple colonies. Dashed line represents average fluorescence of Hoechst nuclear stain. n = 6 NANOG/KLF4/SOX2/nuclei; n = 14 GATA6/OTX2/POU3F1; n = 14 BRACHYURY/FOXA2. BRA, BRACHYURY.

DOI: https://doi.org/10.7554/eLife.32839.002

The following figure supplement is available for figure 1:

**Figure supplement 1.** Micropatterned EpiLC colonies begin as an epithelial monolayer that increases density over time.

DOI: https://doi.org/10.7554/eLife.32839.003

formative state of pluripotency whereby naïve pre-implantation markers, present in the blastocyst, have been downregulated but differentiation has not yet commenced (*Figure 1B*) (*Morgani et al., 2017*; *Smith, 2017*). To establish an in vitro system to model mouse gastrulation, we reasoned that we should start with a PSC population comparable to the in vivo Epi at this time. Global transcriptional profiling identified EpiLCs as the closest in vitro counterpart of the formative Epi (*Hayashi et al., 2011*; *Smith, 2017*; *Kalkan et al., 2017*; *Kalkan and Smith, 2014*). We sought to determine whether this correlation held when EpiLCs were grown on micropatterned surfaces. EpiLCs were generated as described (*Hayashi et al., 2011*) and plated onto 1000 µm diameter micropatterns (*Figure 1C*). Fibronectin and Laminin are basement membrane components present at the Epi-VE interface of peri-gastrulation mouse embryos (*Viotti et al., 2014*; *Bedzhov and Zernicka-Goetz, 2014*). While EpiLCs are generated on a Fibronectin substrate (*Hayashi et al., 2011*), we noted superior adhesion of cells to the micropatterns when coating them with Laminin.

Like the pre-streak Epi, micropatterned EpiLCs expressed the pluripotency-associated markers POU5F1 (OCT4), SOX2, NANOG and OTX2, as well as the post-implantation Epi marker, POU3F1 (OCT6) (*Figure 1B,D*) (*Zhu et al., 2014*; *Acampora et al., 2013*; *Avilion et al., 2003*; *Rosner et al., 1990*; *Hart et al., 2004*). Neither the Epi of the pre-gastrula embryo nor EpiLCs expressed KLF4, a key regulator of naïve pluripotency (*Figure 1B,D*) (*Guo et al., 2009*). The pre-streak Epi does not express lineage-associated markers such as GATA6, FOXA2, CDX2 or BRACHYURY and, these were also absent from EpiLC colonies (*Figure 1B,D*) except in rare cells that we interpret as having spontaneously differentiated.

Micropatterned EpiLC colonies formed an epithelial monolayer (marked by CDH1, also referred to as E-CADHERIN) (*Figure 1E*, *Figure 1—figure supplement 1A*). Cell density within the epithelium increased by nearly a factor of four in 24 hr (*Figure 1—figure supplement 1B*). We noted that micropatterned EpiLCs exhibited a slightly lower cell density than the epithelium of the embryonic Epi at pre- and early gastrulation stages (*Figure 1—figure supplement 1B*). However, in vivo development can also proceed when the Epi cell density is experimentally decreased (*Kojima et al., 2014a*).

The uniform size and circular morphology of micropatterned colonies is amenable to the robust quantification of spatial patterning by measuring protein levels as a function of radial position from the colony center to the colony edge (*Figure 1—figure supplement 1C*) (*Warmflash et al., 2014*; *Etoc et al., 2016*). Such quantification indicated that micropatterned EpiLC colonies were spatially homogeneous (*Figure 1F*), hence we started with a defined population correlating to the pre-gastrulation Epi at approximately E5.5-E6.0.

## EpiLCs exposed to FGF/ACTIVIN/Wnt signaling form a PS-like population

At the onset of gastrulation (E6.25-E6.75), the anterior and posterior of the Epi can be distinguished by the expression of specific markers. SOX2 is elevated within the anterior Epi, while NANOG is restricted to the posterior Epi (*Figure 2A–C*, *Figure 2—figure supplement 1A*) (*Avilion et al., 2003*; *Hart et al., 2004*; *Morkel et al., 2003*; *Osorno et al., 2012*; *Di-Gregorio et al., 2007*). The PS emerges in the proximal posterior Epi and is marked by BRACHYURY expression (*Figure 2A–C*, *Figure 2—figure supplement 1A*) (*Wilkinson et al., 1990*).

When micropatterned EpiLC colonies were cultured with FGF2 and ACTIVIN A for 72 hr, BRACHYURY was induced at the colony periphery (*Figure 2—figure supplement 1B–D*), reminiscent of

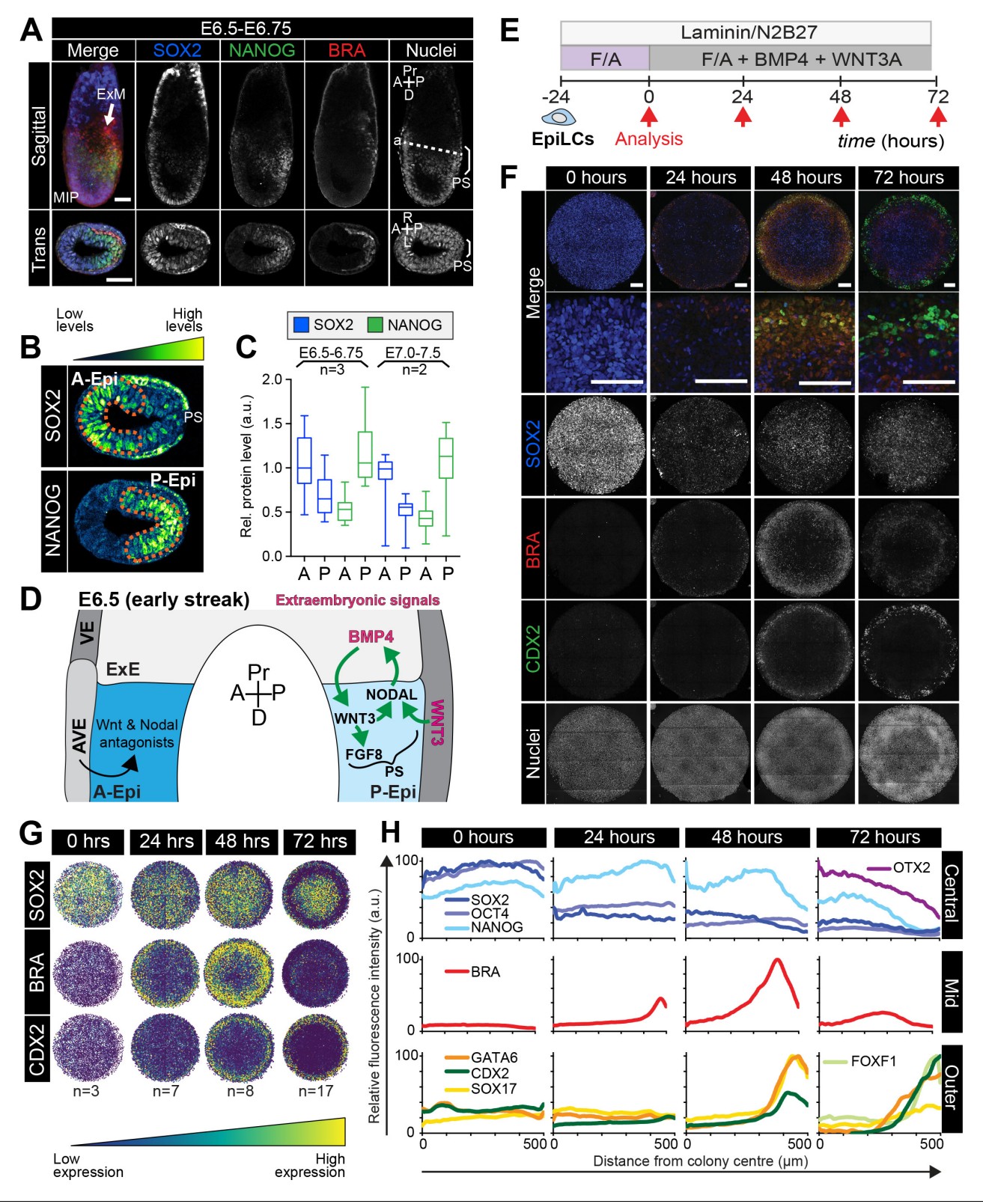

**Figure 2.** Micropatterned EpiLCs undergo spatially organized differentiation. (**A**) Maximum intensity projection (MIP), sagittal and transverse sections of an embryonic day (E) 6.5 mouse embryo. Dashed line marks transverse plane. Non-nuclear anti-BRACHYURY/CDX2/SOX2 VE fluorescence likely represents non-specific binding. ExM, extraembryonic mesoderm; PS, primitive streak; A, anterior; P, posterior; Pr, proximal; D, distal. Scale bars, 50 μm. (**B**) Lookup table (LUT) of SOX2 marking anterior Epi (A-Epi) and NANOG marking posterior Epi (P-Epi). Orange dashed lines delineate regions of
*Figure 2 continued on next page*

*Figure 2 continued*

interest. (**C**) Quantification (5 sections/embryo/ stage) of SOX2 and NANOG in manually selected (panel B) anterior (A) and posterior (P) Epi of E6.5-E6.75 and E7.0-E7.5 embryos, normalized to Hoechst fluorescence. Data depicts mean fluorescence intensity ± S.D. N, number of embryos. No NANOG was observed in the A-Epi hence ~0.5 a.u. equates to background signal. (**D**) BMP, Wnt, Nodal, FGF signaling initiates gastrulation at the P-Epi - extraembryonic ectoderm (ExE) boundary. BMP4 produced by ExE stimulates *Wnt3* expression within proximal Epi. WNT3 produced by Epi and visceral endoderm (VE) triggers *Nodal* and *Fgf8* expression. NODAL promotes *Bmp4* expression in the ExE. The anterior VE (AVE) expresses Wnt and Nodal pathway antagonists, restricting signaling activity to P-Epi. (**E**) EpiLCs were plated onto Laminin-coated micropatterns overnight (−24 hr) in N2B27 with F/A. The following day medium was changed to F/A, BMP4, WNT3A for 72 hr. Colonies were analyzed at 24 hr intervals. (**F**) MIPs of immunostained 1000 μm diameter colonies. All subsequent data represents 1000 μm diameter colonies. Upper two panels represent a merge of the markers shown below. Second panel shows high magnification of colony edge. Scale bars, 100 μm. BRA, BRACHYURY. (**G**) Depiction of average positional marker expression across multiple colonies. Each dot represents a single cell. (**H**) Quantification of voxel fluorescence intensity from colony center (0) to edge (500). Data represents average voxel intensity relative to maximum voxel intensity across time course/marker. For 0,24,48,72 hr respectively, POU5F1/NANOG n = 5,3,3,3, SOX2 n = 15,7,21,20, BRACHYURY n = 11,9,10,12, GATA6/SOX17/CDX2 n = 3,5,6,5. Markers grouped by spatial distribution within colonies. OTX2 and FOXF1 only analyzed at 72 hr.

DOI: https://doi.org/10.7554/eLife.32839.004

The following figure supplements are available for figure 2:

**Figure supplement 1.** FGF, ACTIVIN and endogenous WNT induce a primitive streak state.
DOI: https://doi.org/10.7554/eLife.32839.005

**Figure supplement 2.** Robust micropattern differentiation of EpiLCs.
DOI: https://doi.org/10.7554/eLife.32839.006

PS formation. In the presence of a small molecule inhibitor of Wnt signaling, XAV939 (*Huang et al., 2009*), BRACHYURY expression was abolished and SOX2 was homogeneously expressed at high levels (*Figure 2—figure supplement 1B–D*). BMP inhibition had no obvious effect on BRACHYURY (data not shown). Hence in micropatterned colonies, as in EpiSC cultures (*Kim et al., 2013*; *Kurek et al., 2015*; *Tsakiridis et al., 2014*; *Wu et al., 2015*; *Sumi et al., 2013*), BRACHYURY expression was dependent on endogenous Wnt signaling. This suggested that FGF and ACTIVIN support a SOX2-high anterior Epi-like state, while FGF and ACTIVIN combined with endogenous WNT trigger a reduction in SOX2 levels, as in the posterior Epi, and PS gene expression at the colony periphery (*Figure 2—figure supplement 1E*, *Table 1*). The later germ layer markers GATA6, SOX17 and CDX2 (*Figure 2—figure supplement 1C*) were not expressed under these conditions indicating that additional factors were required to stimulate further differentiation.

**Table 1.** Summary of cell fates arising in the presence of various signaling factors.

Cell fates generated after 72 hr of mouse micropattern differentiation with described cytokine combinations. It should be noted that cells do not detect these signals homogeneously. WNT inhibition (WNTi) refers to XAV treatment; ACTIVIN inhibition (ACTIVINi) refers to the absence of ACTIVIN and SB431542.

| Signaling pathways | Outcome |
|---|---|
| FGF, ACTIVIN, (WNTi) | Epiblast |
| FGF, ACTIVIN, endogenous WNT | Epiblast<br>PS |
| FGF, ACTIVIN, BMP, WNT | Posterior epiblast<br>PS<br>Extraembryonic mesoderm<br>Embryonic mesoderm |
| FGF, ACTIVIN, WNT | Anterior epiblast<br>Definitive endoderm<br>Axial mesoderm or<br>Anterior PS |
| FGF, BMP, WNT (ACTIVINi) | Epiblast |

DOI: https://doi.org/10.7554/eLife.32839.007

## Posteriorization of EpiLCs after 24 hr exposure to FGF/ACTIVIN/WNT/BMP

In vivo, gastrulation is triggered by a combination of signals from both embryonic Epi cells and the extraembryonic tissues that lie adjacent to the proximal posterior Epi. The extraembryonic signals include WNT3 produced by the VE, and BMP4 produced by the extraembryonic ectoderm (ExE) (*Tam and Loebel, 2007*; *Robertson, 2014*) (*Figure 2D*). For the mouse micropattern differentiation we utilized pluripotent EpiLCs, corresponding to the Epi cells of the embryo, and thus the system likely lacked the neighboring extraembryonic cell types and the signals that they provide. We therefore asked whether supplying these signals exogenously could mirror the in vivo signaling environment and initiate gastrulation-like events in vitro.

EpiLCs were plated overnight onto micropatterns in defined serum-free medium containing 12 ng/ml FGF2 and 20 ng/ml ACTIVIN A, supplemented the following day with 50 ng/ml BMP4 and 200 ng/ml WNT3A (*Figure 2E*). Under these conditions, EpiLCs underwent robust and reproducible organized germ layer specification (*Figure 2F–H*, *Figure 2—figure supplement 2A–C*).

After 24 hr of differentiation, the EpiLC micropatterned colonies gave rise to two populations – a central population expressing the Epi markers POU5F1, SOX2 and NANOG, and an outer population expressing the PS marker BRACHYURY (*Figure 2F,G*). From 0 to 24 hr, SOX2 levels were reduced approximately 2-fold within the central population (*Figure 2F–H*, *Figure 2—figure supplement 2D*). Conversely, there was no change in NANOG expression (*Figure 2H*, *Figure 2—figure supplement 2D*). Hence, a NANOG-positive, SOX2-low state emerged as is present in the posterior Epi of the embryo (*Hoffman et al., 2013*) (*Figure 2A–C*, *Figure 2—figure supplement 2D*). Concomitantly, BRACHYURY was induced at the colony edge (*Figure 2F–H*). BRACHYURY and SOX2 expression was predominantly mutually exclusive but showed a degree of overlap (*Figure 2—figure supplement 2E*), which was also observed in cells within the PS in vivo (*Figure 2A*) and may mark an Epi-PS transition state. At this time, no later germ layer-associated markers (GATA6, SOX17, CDX2, FOXA2) were expressed (*Figure 2F,H*). Hence, the first 24 hr of micropattern differentiation with exogenous BMP and WNT generated populations resembling the posterior Epi and emerging PS of the mouse embryo at approximately E6.25-E6.75, as observed in the presence of FGF, ACTIVIN and endogenous WNT (*Figure 2—figure supplement 1B–D*).

## Identification of marker signatures to track germ layer differentiation

As gastrulation proceeds, an increasingly complex array of populations are specified – the anterior and posterior Epi, the PS, the embryonic and extraembryonic mesoderm (arising from the posterior PS) and the DE and AxM (arising from the anterior PS) (*Figure 3—figure supplement 1A*). To identify these cell states in vitro in the absence of the spatial and temporal context of the embryo, we sought to establish marker signatures of the cell types present in gastrula stage embryos. To do this, we performed immunostaining for a panel of factors on gastrulating wild-type mouse embryos, and supplemented our observations with published data (*Supplementary file 1* and *2*).

At E6.5-E7.5, the anterior Epi expresses high levels of SOX2 and OTX2 while the posterior Epi expresses low levels of SOX2 and high levels of NANOG (*Figure 2A–C*, *Figure 3—figure supplement 1B*). In agreement with a recent spatial transcriptional analysis of gastrulating mouse embryos (*Peng et al., 2016*), our immunostaining data suggested that posterior Epi cells may also express low levels of OTX2 (*Figure 3—figure supplement 1B*). At E7.0-E7.5, a fraction of distal posterior Epi cells begin to express FOXA2 (*Figure 3A*) (*Burtscher and Lickert, 2009*), some of which also express OTX2 (*Peng et al., 2016*; *Engert et al., 2013*).Then, by E7.75, CDX2 is expressed throughout the posterior Epi (*Figure 3B*) (*Deschamps and van Nes, 2005*). By E7.75-E8.0, SOX2 continues to be expressed at high levels in the anterior neurectoderm and at low levels in the posterior, while NANOG is no longer observed within the Epi (see *Supplementary file 2*) (*Osorno et al., 2012*).

Throughout gastrulation, BRACHYURY is expressed by cells within the PS (*Figures 2A* and *3A, B*, *Figure 3—figure supplement 1C,F*). The first cells to leave the posterior Epi and exit the PS at E6.5-E6.75 coexpress BRACHYURY and GATA6 (*Figure 3—figure supplement 1C*). Over time, these cells adopt distinct mesodermal and DE identities. Cells that exit the posterior PS and move proximally into the extraembryonic region generate the extraembryonic mesoderm. The extraembryonic mesoderm forms structures involved in the exchange of materials between the embryo and the mother, including the allantois and yolk sac. Additionally, the extraembryonic mesoderm is a source

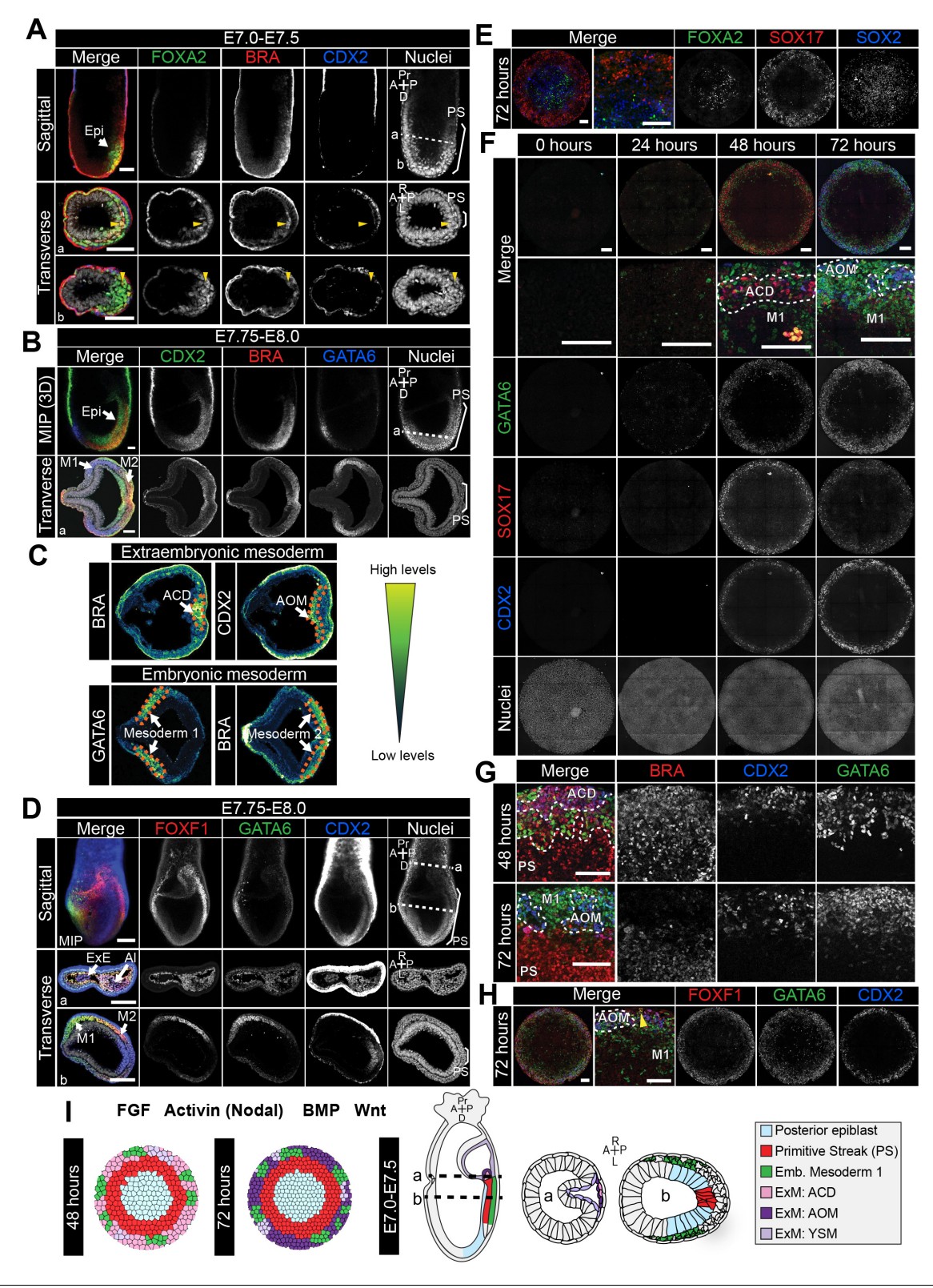

**Figure 3.** Assignment of cell identities to micropattern-differentiated EpiLC populations. (**A,B,D**) Confocal maximum intensity projections (MIP), sagittal optical sections and transverse cryosections of immunostained gastrulating embryos. Dashed lines mark transverse plane. Epi, epiblast; PS, primitive streak; M1, Mesoderm1; M2, Mesoderm2; ACD, allantois core domain; AOM, allantois outer mesenchyme; ExE, extraembryonic ectoderm; Al, allantois; ExM, extraembryonic mesoderm; A, anterior; P, posterior; Pr, proximal; D, distal; R, right; L, left. Scale bars, 50 µm. (**A**) Yellow arrowheads mark

*Figure 3 continued on next page*

*Figure 3 continued*

BRACHYURY/FOXA2-coexpressing cells within the anterior PS. (**C**) LUT of immunostaining of BRACHYURY marking extraembryonic mesoderm allantois core domain (ACD) and CDX2 expressed highly in allantois outer mesenchyme (AOM) (upper panels) as well as GATA6 marking anteriorly migrated embryonic mesoderm (Mesoderm 1) and BRACHYURY marking embryonic mesoderm close to the PS (Mesoderm 2) (lower panels). Orange dashed lines delineate regions of interest. (**E,F,H**) MIPs of immunostained micropatterns. High magnification shows region at the colony edge. Yellow arrowhead H marks GATA6/FOXF1 cell. Scale bars, 100 µm. (**G**) High magnification of colony edge. Outer domain represents a mixture of populations often organized in clusters, highlighted by dashed lines. At 48 hr the ACD population coexpressed BRACHYURY and CDX2, M1 expressed GATA6. By 72 hr, outer cells expressed CDX2 (AOM) or GATA6 (M1). BRACHYURY marked PS cells. (**I**) Schematic diagram summarizing the cell identities observed at 48 and 72 hr of in vitro differentiation, under conditions described in B and corresponding in vivo fates. Dashed lines mark transverse plane. ExM, extraembryonic mesoderm.

DOI: https://doi.org/10.7554/eLife.32839.008

The following figure supplements are available for figure 3:

**Figure supplement 1.** Populations arising during in vivo and in vitro differentiation.
DOI: https://doi.org/10.7554/eLife.32839.009
**Figure supplement 2.** Micropatterns in the presence of BMP express SOX17 and *Hhex* but not *Gsc*.
DOI: https://doi.org/10.7554/eLife.32839.010

of hematopoietic progenitors and factors associated with early hematopoiesis, such as *Sox17* and *Hhex*, are expressed within the allantois (*Thomas et al., 1998*; *Bedford et al., 1993*; *Sakamoto et al., 2007*; *Burtscher et al., 2012*). At E7.75-E8.0, the extraembryonic mesoderm can be subdivided into the allantois core domain (ACD) - expressing BRACHYURY, SOX17, CDX2, FOXF1, the allantois outer mesenchyme (AOM) - expressing SOX17, CDX2, GATA6, FOXF1 and the yolk sac mesoderm (YSM) - expressing GATA6 and FOXF1 (*Figure 3C,D*, *Figure 3—figure supplement 1D*) (*Deschamps and van Nes, 2005*; *Burtscher et al., 2012*; *Freyer et al., 2015*; *Choi et al., 2012*; *Peterson et al., 1997*; *Fleury et al., 2015*). The allantois is reported to express *Gata6* at this time (*Morrisey et al., 1996*), although at the protein level GATA6 was not evident until slightly later stages. At E8.5, CDX2, SOX17 and GATA6 are expressed throughout the allantois (*Supplementary file 2*). Over time, cells of the ACD contribute to the AOM to support allantois elongation (*Downs et al., 2009*).

Cells that originate from the posterior PS and move in an anterior direction around the embryo will form the embryonic mesoderm. At E7.5-E8.0, based on protein expression, we could distinguish two populations of embryonic mesoderm, which we refer to here as Mesoderm 1 and Mesoderm 2 (*Figure 3C*). Mesoderm 1 cells, which exited the PS earlier and were located more anteriorly, expressed GATA6 and OTX2 (*Figure 3B–D*, *Figure 3—figure supplement 1B*) (*Freyer et al., 2015*; *Morrisey et al., 1996*). Mesoderm 2 cells, which left the PS later and so were more posterior, expressed BRACHYURY and FOXF1 (*Figure 3B–D*) (*Peterson et al., 1997*). To note, GATA6 and FOXF1 showed a degree of overlap within the region between Mesoderm 1 and 2 (*Figure 3D*).

Over time, the PS extends distally within the cup-shaped mouse embryo and cells that emanate from the anterior PS give rise to the DE and AxM. Although BRACHYURY is expressed along the length of the PS, at the anterior PS a fraction of cells coexpress BRACHYURY and FOXA2 while others express only FOXA2 (*Figure 3A*) (*Burtscher and Lickert, 2009*). Both DE and AxM cells express FOXA2, OTX2, *Gsc* and *Hhex*, although *Gsc* and *Hhex* may be present only transiently within the AxM (*Figure 3A*, *Figure 3—figure supplement 1E*) (*Thomas et al., 1998*; *Wu et al., 2017*; *Rodriguez et al., 2001*; *Belo et al., 1998*; *Sasaki and Hogan, 1993*). Additionally, DE cells express SOX17 (*Figure 3—figure supplement 1E*) and GATA6 (*Viotti et al., 2014*; *Freyer et al., 2015*). AxM cells also express BRACHYURY (*Figure 3—figure supplement 1F*) (*Wilkinson et al., 1990*).

Utilizing these in vivo marker signatures (*Supplementary file 1*), we sought to assign identities to the cell populations arising at 48–72 hr of micropattern differentiation, and consequently map the in vitro differentiation to in vivo development.

## FGF/ACTIVIN/WNT/BMP triggers spatially organized posterior fate specification

The first 24 hr of micropattern differentiation in the presence of FGF, ACTIVIN, BMP and WNT resulted in EpiLC posteriorization and the emergence of a PS-like population (*Figure 2F,G*). At 48 and 72 hr of micropattern differentiation, further spatially organized germ layer differentiation was

observed. Three colony domains (central, mid and outer concentric circles) were evident based on marker expression (*Figure 2—figure supplement 2A–C*) and the populations within these domains were largely conserved between time points.

At both 48 and 72 hr, the colony center continued to express NANOG and low levels of POU5F1 and SOX2 as in the posterior Epi (*Figure 2F–H*, *Figure 2—figure supplement 2D*). Additionally, at 72 hr, OTX2 and FOXA2 were observed within the colony center (*Figure 3E*, *Figure 3—figure supplement 1G*) analogous to the expression of these markers within the mid to distal region of the posterior Epi at E7.0-E7.5 (*Figure 3A*) (*Engert et al., 2013*). While OTX2 and SOX2 are also highly expressed within the in vivo anterior Epi (*Figure 3—figure supplement 1B*), the additional expression of the posterior-restricted marker NANOG, as well as FOXA2, within the colony center (*Figure 2A*, *Figure 2—figure supplement 2D*, *Figure 3A,E*) suggested that these cells were more similar to posterior than anterior Epi. In vivo, CDX2 expression is induced within the posterior Epi at E7.5-E7.75 (*Figure 3B*) but was not observed within the center of micropatterned colonies (*Figures 2F* and *3F*). Therefore, based on the expression of FOXA2 but not CDX2, the central population likely correlated to the posterior Epi later than E7.0, but prior to E7.75.

At 24 hr, BRACHYURY marked a PS-like population at the colony periphery (*Figure 2F–H*). However, at 48 and 72 hr BRACHYURY-positive cells were observed more centrally, within the mid micropattern domain (*Figure 2F–H*). This inwards shift of BRACHYURY suggested either a wave of gene expression propagating throughout the colony or an inward movement of BRACHYURY-expressing cells. In vivo, BRACHYURY is expressed by cells within the PS, but is additionally present in the first cells emanating from the PS – where it is coexpressed with GATA6, and in the extraembryonic mesoderm ACD – where it is coexpressed with CDX2 and SOX17. While we observed cells that coexpressed BRACHYURY and GATA6 or CDX2/SOX17 (discussed later), the majority of BRACHYURY-positive cells did not express these markers, and hence likely corresponded to PS (*Figure 3—figure supplement 1H,I*).

The outermost micropattern domain comprised several distinct cell populations with expression signatures reminiscent of embryonic and extraembryonic mesoderm. A small fraction of cells coexpressed GATA6 and BRACHYURY, as in the first cells leaving the PS (*Figure 3—figure supplement 1C,J,K*). Additionally, we observed cells that coexpressed GATA6 and OTX2, as in Mesoderm 1 (*Figure 3B*, *Figure 3—figure supplement 1B,G*). While in vivo we could also discern a second population of embryonic mesoderm (Mesoderm 2) that expressed FOXF1 and BRACHYURY (*Figure 3B–D*), within the micropatterned colonies FOXF1 was restricted to the colony periphery in a spatially distinct domain from BRACHYURY (*Figures 2H* and *3H*). Hence Mesoderm 2 was likely not generated under these in vitro differentiation conditions.

Within the same outer colony domain, we identified populations resembling extraembryonic mesoderm cell types. CDX2 was expressed within the outer domain from 48 hr of differentiation (*Figures 2F–H* and *3F*). At this time, almost all CDX2-positive cells coexpressed BRACHYURY and SOX17 (*Figures 2F* and *3F,G*, *Figure 3—figure supplement 1*). In vivo, coexpression of BRACHYURY, SOX17 and CDX2 is first observed within extraembryonic mesoderm cells of the ACD (*Figure 3—figure supplement 1D*). While BRACHYURY, SOX17 and CDX2 are also all expressed within cells of the hindgut at later stages of development (E8.5) (*Lewis and Tam, 2006*), this micropattern population did not express additional hindgut markers such as FOXA2 (*Monaghan et al., 1993*; *Ang et al., 1993*; *Ruiz i Altaba et al., 1993*) (*Figure 3E*). Hence BRACHYURY/SOX17/CDX2-positive cells correlated most strongly to the extraembryonic mesoderm ACD.

At 72 hr, the BRACHYURY/SOX17/CDX2 population was no longer observed. CDX2 and SOX17 continued to be coexpressed (*Figure 3F,G*) but these cells now lacked BRACHYURY expression (*Figures 2F* and *3F,G*, *Figure 3—figure supplement 1J*). At 72 hr, CDX2-positive cells also expressed FOXF1, another marker found within the extraembryonic mesoderm (*Figure 3D*). Hence, SOX17/CDX2/FOXF1-positive cells likely corresponded to the AOM, suggesting a temporal progression of ACD cells to an AOM state, as in vivo. We also observed a rarer population of cells that coexpressed FOXF1 and GATA6 (*Figure 3H*, yellow arrowhead) as in the YSM or embryonic mesoderm positioned between Mesoderm 1 and 2 (*Figure 3D*). While both embryonic (GATA6) and extraembryonic (SOX17/CDX2/FOXF1) mesoderm-like populations were present within the outer micropattern domain, they tended to exist within discrete clusters (*Figure 3G*).

During the micropattern differentiation, multiple DE-associated markers were expressed, namely FOXA2, SOX17, GATA6 and OTX2. In vivo these markers are coexpressed within DE cells (*Figure 3—*

*figure supplement 1E*) while in the micropattern differentiation FOXA2, SOX17 and GATA6 were expressed in a mostly mutually exclusive manner (*Figure 3E*, *Figure 3—figure supplement 2A*), hence they marked separate non-DE populations. Furthermore, BRACHYURY and FOXA2 were expressed within distinct micropattern domains (mid and central respectively) (*Figures 2F,G* and *3E*) suggesting that AxM cells were not present. To further validate these conclusions, we assessed the expression of the DE and AxM markers *Gsc* and *Hhex* using a $Gsc^{GFP/+}$; $Hhex^{RedStar/+}$ dual reporter ESC line (*Villegas et al., 2013*). After 72 hr of differentiation, $Hhex^{RedStar}$ was observed at the outer colony edge but $Gsc^{GFP}$ was not expressed (*Figure 3—figure supplement 2B*). The expression of $Hhex^{RedStar}$ and SOX17, without FOXA2 and $Gsc^{GFP}$ expression, confirmed the absence of DE and AxM fates and likely indicated the presence of hematopoietic progenitors that arise from the allantois (*Thomas et al., 1998*; *Crompton et al., 1992*) and

Therefore, in vitro, a combination of BMP, WNT, ACTIVIN (NODAL), and FGF promoted the specification and spatial organization of posterior Epi (center), PS (mid) and embryonic and extraembryonic mesoderm (outer), recapitulating gastrulation events occurring within the posterior of the mouse gastrula (*Figure 3I*, *Table 1*). However, fates arising from the anterior PS including DE and AxM were not formed under these conditions.

## Micropattern differentiation involves a TGFβ-regulated EMT

One of the primary hallmarks of gastrulation is an EMT, involving downregulation of the epithelial marker CDH1 (E-CADHERIN) and upregulation of the mesenchymal marker CDH2 (N-CADHERIN) in cells ingressing through the PS (*Figure 4A,B*). Epi cells that do not undergo an EMT differentiate into neurectoderm, while those that undergo an EMT emanate from the PS and acquire mesoderm or DE identities (*Arnold and Robertson, 2009*; *Ferrer-Vaquer et al., 2010*). We asked whether micropattern differentiation engaged these same morphogenetic processes.

A PS-like population arose after 24 hr of in vitro differentiation (*Figure 2F*) followed by the formation of a 2–3 cell layer ridge at the colony perimeter at 48 hr (*Figure 4C,D*). By 72 hr, the ridge was positioned more centrally, suggesting an inwards movement, resulting in a volcano-like structure (*Figure 4C,D*). Initially, the ridge overlapped with the BRACHYURY/CDX2 coexpression domain but, at 72 hr, was positioned at the border between the BRACHYURY-positive PS and CDX2-positive AOM populations (*Figure 4—figure supplement 1A*). Cells at the border of the BRACHYURY-expressing region downregulated the epithelial marker CDH1 and upregulated the mesenchymal marker CDH2 (*Figure 4D,E*, *Figure 4—figure supplement 1B*). As in vivo, the outer CDH2 expression domain correlated with the position of the PS (BRACHYURY), embryonic mesoderm (GATA6) and extraembryonic mesoderm (CDX2) populations (*Figure 4F*). Furthermore, both the intermediate PS-like domain and the outer embryonic and extraembryonic mesoderm domain expressed SNAIL (*Figure 4—figure supplement 1C–E*), a transcriptional repressor that regulates the gastrulation EMT (*Carver et al., 2001*). At 48 hr, CDH2-positive cells emerged at the base of the colony, beneath the CDH1-positive epithelial layer, and were observed more centrally over time (*Figure 4D,E*). We also occasionally observed BRACHYURY-expressing cells in more central positions at 72 hr (*Figure 4F*), which could suggest an inwards migration of mesenchymal PS derivatives between the upper epithelium and the surface of the micropattern slide. Conversely, central posterior Epi-like cells, maintained CDH1 (*Figure 4D,E*).

Various signaling pathways including Wnt, FGF and TGFβ regulate EMT in development and cancer (*Ferrer-Vaquer et al., 2010*; *Kang and Massagué, 2004*; *Ciruna and Rossant, 2001*). In particular, the role of TGFβ signaling through SMAD2/3 has been well characterized (*Xu et al., 2009*). Mice with null mutations in *Smad2/3* or *Nodal* do not gastrulate and lack normal mesoderm structures (*Dunn et al., 2004*; *Nomura and Li, 1998*; *Weinstein et al., 1998*; *Conlon et al., 1994*). To determine whether SMAD2/3 signaling regulated EMT in the in vitro micropattern system, we cultured EpiLC micropatterned colonies in medium containing FGF2, BMP4, WNT3A but lacking ACTIVIN A and supplemented with a small molecule inhibitor of the ALK5 receptor, SB431542 (referred to as ACTIVINi). In the absence of Activin (Nodal) signaling, cells maintained high levels of CDH1 and accumulated at the edge of colonies (*Figure 4G,H*, *Figure 4—figure supplement 1F*). Furthermore, they failed to downregulate SOX2 and did not differentiate, evidenced by the lack of BRACHYURY, GATA6, CDX2 or SOX17 expression (*Figure 4I*, *Figure 4—figure supplement 1F*). Thus, in these flat-disc-shaped micropatterns, SMAD2/3 signaling regulated the EMT associated with an exit from pluripotency and onset of differentiation confirming that in vitro micropattern differentiation and in

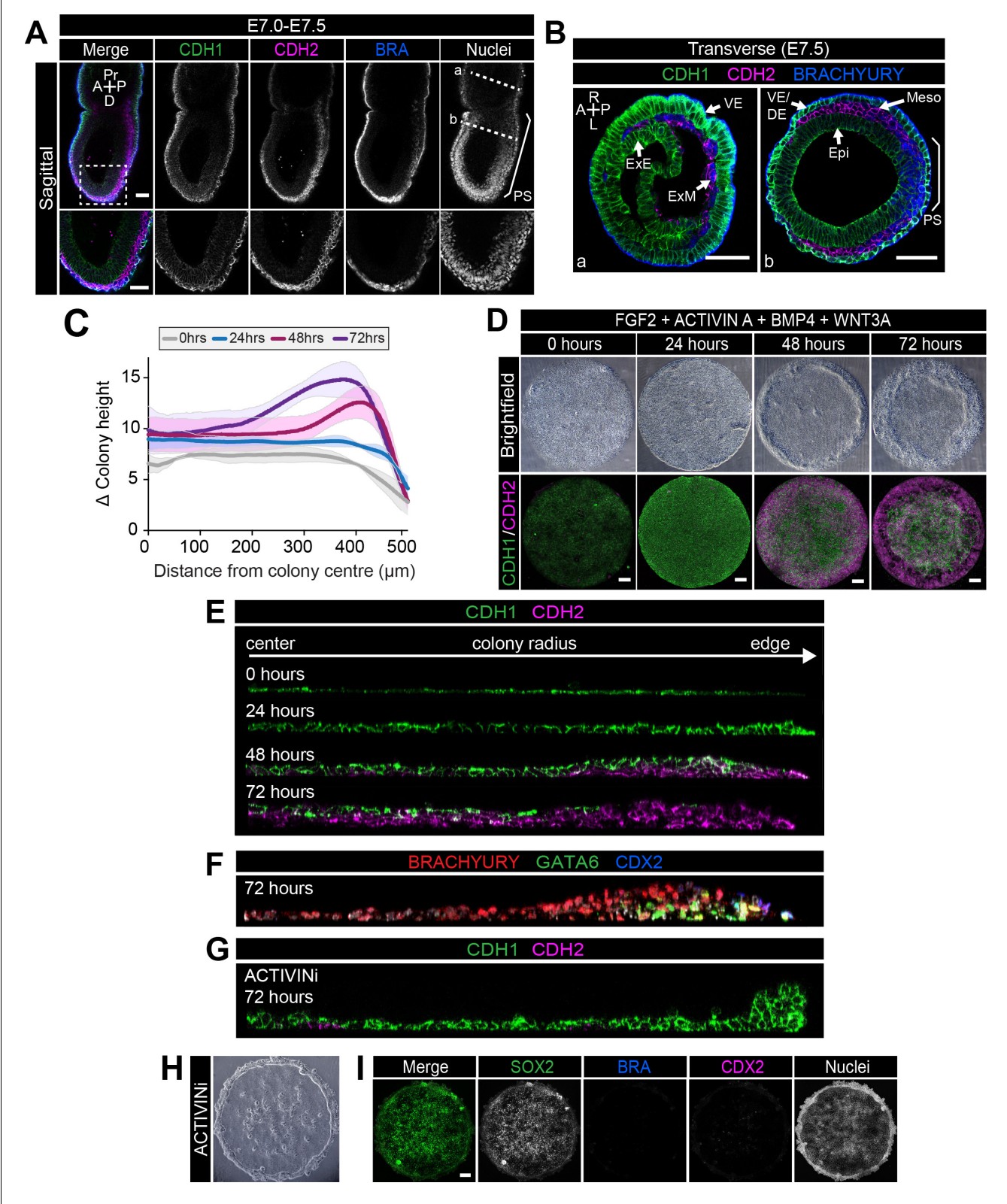

**Figure 4.** EMT is associated with micropatterned EpiLC differentiation. Data from colonies differentiated as in *Figure 2E*. (A,B) Sagittal (A) and transverse sections (B) of late streak embryo. Dashed box marks high magnification region in lower panel. Dashed lines mark transverse planes in B. Non-nuclear anti-BRACHYURY VE fluorescence represents non-specific binding. A, anterior; P, posterior; Pr, proximal; D, distal; L, left; R, right; VE/DE, visceral endoderm/definitive endoderm; ExE, extraembryonic ectoderm; ExM, extraembryonic mesoderm; Epi, epiblast; Meso, mesoderm. Scale bars,

*Figure 4 continued on next page*

*Figure 4 continued*

50 μm. (**C**) Quantification of colony height from colony center (0) to edge (500) across multiple colonies, three independent experiments, 0 hr: n = 11, 24 hr: n = 15, 48 hr: n = 17, 72 hr: n = 18. (**D**) Time-course showing brightfield images (upper panels) and MIPs of comparable immunostained colonies (lower panels). Scale bars, 100 μm. (**E–G**) Images of z-axis profile from colony center (left) to edge (right). (**G–I**) EpiLCs were plated onto micropatterns overnight with F/A. The following day medium was changed to F/A, BMP4, WNT3A (**E,F**) or medium blocking Activin/Nodal signaling - FGF2, BMP4, WNT3A, SB431542 (ACTIVINi, (**G–I**). (**H**) brightfield image of ACTIVINi colony. (**I**) MIPs of immunostained ACTIVINi colonies at 72 hr differentiation. Scale bars, 100 μm. BRA, BRACHYURY.

DOI: https://doi.org/10.7554/eLife.32839.011

The following figure supplement is available for figure 4:

**Figure supplement 1.** Cells undergo an epithelial to mesenchymal transition during gastrulation and in vitro differentiation.

DOI: https://doi.org/10.7554/eLife.32839.012

vivo gastrulation are regulated by common pathways and processes even though their geometries (flat-disc versus cup-shaped) are distinct.

## Colony diameter is a critical factor involved in patterning

It was previously shown that, in the micropattern system, hESCs give rise to the broadest spectrum of cell fates when differentiated within colonies of 500–1000 μm diameter (*Warmflash et al., 2014*; *Etoc et al., 2016*). When the micropattern diameter was decreased, the outer cell fate domains were preserved while the inner cell fates were lost, suggesting an edge-sensing input into the differentiation (*Warmflash et al., 2014*; *Etoc et al., 2016*). We asked whether colony diameter also affected cell fate specification and patterning of mouse PSCs.

At 72 hr of mouse PSC differentiation on micropatterns of 1000 μm diameter, three concentric domains (Regions A-C) could be defined with respect to SOX2, BRACHYURY and CDX2 expression (*Figures 2F* and *5A*). The most central domain was Epi-like and predominantly expressed SOX2 (Region A), the mid domain was PS-like and predominantly expressed BRACHYURY (Region B), and the outermost region (Region C) comprised both CDX2-positive extraembryonic mesoderm and GATA6-positive embryonic mesoderm cells (*Figure 5A*). While SOX2 levels were highest within the colony center, it was expressed at reduced levels throughout all domains (*Figure 5A,B*).

We noted that the temporal order of differentiation was maintained across the different colony diameters analyzed - 500, 225, 140 and 80 μm (*Figure 5C–E*). BRACHYURY-expressing PS-like cells were observed at 24 hr, followed by BRACHYURY/CDX2 coexpression within ACD-like cells at 48 hr and the emergence of cells expressing CDX2 but not BRACHYURY, as in the AOM, at 72 hr (*Figure 5C–D*). However, the spatial organization of cell fates after 72 hr of differentiation was dependent on colony diameter (*Figure 5B*). Within 500 μm diameter colonies, the outer extraembryonic mesoderm population marked by CDX2 was maintained but, in contrast to colonies of 1000 μm diameter, BRACHYURY-positive PS cells were positioned within the colony center in place of the SOX2 only Epi-like population (*Figure 5B–E*, *Figure 5—figure supplement 1*).

At even smaller micropattern diameters of 80–140 μm, colonies were comprised predominantly of CDX2 or SOX2 –positive cells with almost no BRACHYURY expression observed (*Figure 5B–E*, *Figure 5—figure supplement 1*). CDX2 and SOX2 marked distinct, apparently randomly positioned, clusters of cells (*Figure 5B–E*, *Figure 5—figure supplement 1*). Within 1000 μm diameter colonies, the outer micropattern domain (Region C) was comprised of both embryonic (GATA6) and extraembryonic (SOX17/CDX2) mesoderm populations (*Figure 3F–H*). Both embryonic and extraembryonic mesoderm populations were also observed within smaller diameter colonies whereby CDX2 and SOX17 -positive cells were present within distinct domains from cells that expressed GATA6 (*Figure 5F*). It should be noted that, on smaller diameter micropatterns, the width to height ratio of colonies was altered such that 80–140 μm diameter colonies generated taller, embryoid body-like aggregates. Over time, these three-dimensional structures exhibited morphological asymmetries (*Figure 5F*), which may explain the loss of radial symmetry in marker expression. Colonies of 225 μm diameter patterned cell fates in a manner intermediate to that observed in colonies of 500 μm and 80–140 μm diameter, with some BRACHYURY cells still observed (*Figure 5C–E*, *Figure 5—figure supplement 1*).

Taken together, these data show that micropattern diameter is a critical factor in determining mouse PSC fate specification and patterning. While the non-uniform, three-dimensional geometry of

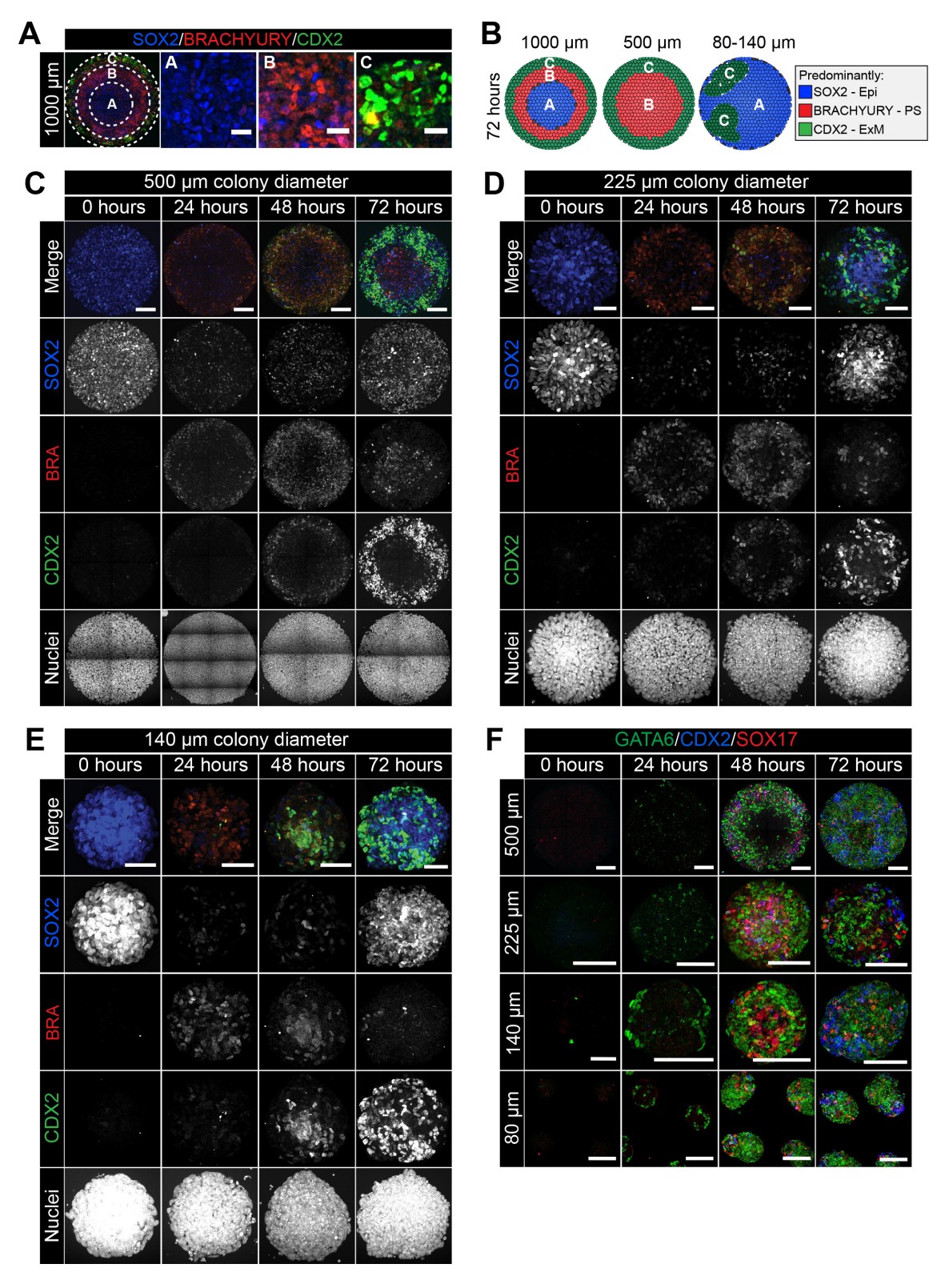

**Figure 5.** Smaller diameter colonies pattern in the same order of events but lose central populations. (**A**) EpiLCs were differentiated with FGF2 and ACTIVIN A (F/A), BMP4 and WNT3A as described in *Figure 2E*. Confocal optical section of a representative 1000 µm diameter colony after differentiation. Dashed circles define 3 regions of distinct marker expression, shown at higher magnification in adjacent panels. While SOX2 is expressed quite broadly, regions were defined based on the marker that was predominantly expressed. Region A (central) = SOX2 (blue), Region B

*Figure 5 continued on next page*

*Figure 5 continued*

(intermediate) = BRACHYURY (red), Region C (outer) = CDX2 (green). Scale bars, 25 µm. (**B**) Schematic diagram showing the changing marker expression in colonies of different diameters. (**C–F**) Representative confocal maximum intensity projections of colonies at 0, 24, 48 and 72 hr after addition of BMP4 and WNT3A to F/A medium. Images show colonies of 500 µm, 225 µm, 140 µm and 80 µm diameter. Scale bars, 100 µm.

DOI: https://doi.org/10.7554/eLife.32839.013

The following figure supplement is available for figure 5:

**Figure supplement 1.** Patterning of cell fates is altered at different colony diameters.

DOI: https://doi.org/10.7554/eLife.32839.014

smaller colonies made data difficult to interpret, the loss of central populations within 500 µm diameter colonies indicates that, like human PSCs (*Warmflash et al., 2014*), mouse PSCs may specify fates as a function of distance from the colony edge.

## Micropattern colonies exhibit position-dependent BMP signaling

While the cell culture medium provided homogeneous signals to the micropatterned colonies, different cell fates emerged within distinct radial domains. To determine whether this patterning correlated to a position-dependent interpretation of signals, we focused on BMP, a key upstream signal necessary for gastrulation with an effective antibody readout of activity - nuclear localization of phosphorylated SMAD1/5/8 (pSMAD1/5/8). In vivo, at the early streak stage (E6.5-E6.75), BMP4 is expressed by the ExE and later (E7.5-E8.0) by the allantois and chorion (*Lawson et al., 1999*) (*Figure 6—figure supplement 1A*), and acts on adjacent tissues.

In E6.5-E6.75 embryos, BMP signaling (marked by pSMAD1/5/8) was active at low levels within the proximal, but not distal, Epi and elevated within cells of the PS and embryonic and extraembryonic mesoderm (*Figure 6—figure supplement 1B*). At this stage, pSMAD1/5/8 levels correlated with BRACHYURY expression (*Figure 6A,B*, *Figure 6—figure supplement 1B,C*). From E7.0 onwards, as the PS extended, pSMAD1/5/8 was observed within the posterior PS but not anterior PS, consistent with anterior cells being positioned furthest from the ExE source of BMP4 (*Figure 6—figure supplement 1D,E*). Furthermore, pSMAD1/5/8 was observed in embryonic Mesoderm 1 but not BRACHYURY-positive Mesoderm 2 cells and consequently, the correlation with BRACHYURY expression was lost (*Figure 6—figure supplement 1E–G*).

At 0 hr of micropattern differentiation, nuclear pSMAD1/5/8 was observed at low levels throughout colonies (*Figure 6C*), corresponding to the low BMP signaling activity within the proximal embryonic Epi at E6.5-E6.75 (*Figure 6—figure supplement 1B*). From 24–72 hr of micropattern differentiation, nuclear pSMAD1/5/8 was elevated at the colony edge within the PS, embryonic and extraembryonic mesoderm cell fate domains (*Figure 6C–E*). At 24 and 48 hr, the majority of pSMAD1/5/8-positive cells expressed BRACHYURY but by 72 hr, the fraction of BRACHYURY/pSMAD1/5/8 -positive cells was significantly reduced (*Figure 6C*, *Figure 6—figure supplement 1H, I*). This likely corresponded to the presence of nuclear pSMAD1/5/8, but not BRACHYURY, within Mesoderm 1 cells of the embryo (*Figure 6—figure supplement 1E,F*).

These data revealed that, as with micropattern differentiated hESCs (*Warmflash et al., 2014*; *Tewary et al., 2017*; *Etoc et al., 2016*), signaling activity exhibits radial dependence. Furthermore, the cell types identified within the in vitro micropattern system experienced a comparable BMP signaling history to their in vivo counterparts, with low BMP signaling activity present within posterior Epi-like cells and elevated activity within the posterior PS, embryonic Mesoderm one and extraembryonic mesoderm populations. In vivo, the distal Epi and anterior PS were devoid of BMP activity (*Figure 6—figure supplement 1B,D,E*) but, in the presence of FGF, ACTIVIN, BMP and WNT, a comparable signaling niche that lacked BMP activity was not observed within the micropatterned colonies.

The spatial organization of hESC-derived cell fates during micropattern differentiation is mediated by a combination of receptor occlusion at the colony center and loss of secreted inhibitors from the colony edge (*Tewary et al., 2017*; *Etoc et al., 2016*). To test for the involvement of receptors in the micropattern organization of mouse cell fates, we substituted exogenous WNT3A with a GSK3 inhibitor, CHIR99021 (CHIR), which circumvents the receptor to activate downstream Wnt pathway components (*Figure 6—figure supplement 2A,B*). Under these conditions, CDX2, GATA6

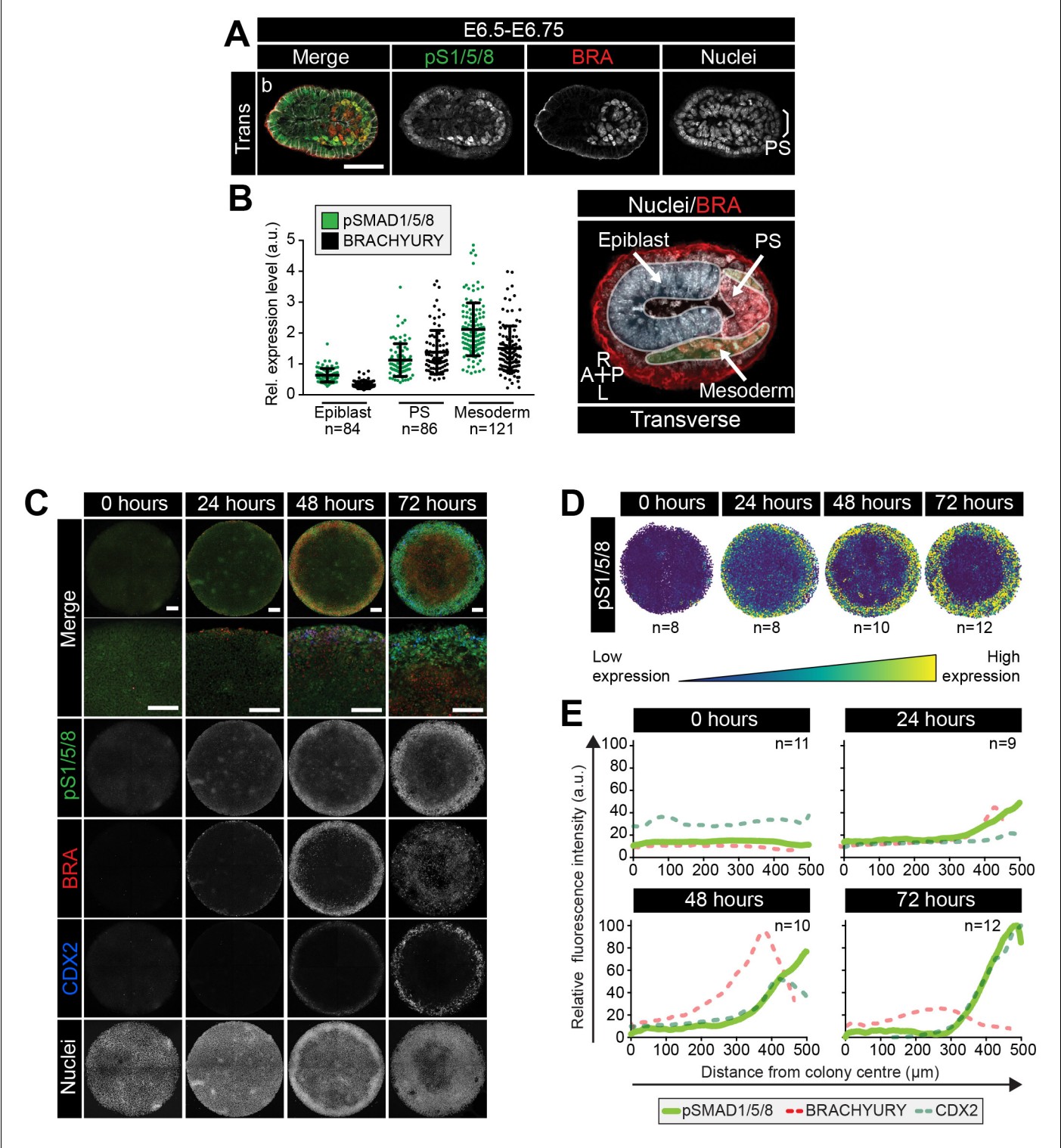

**Figure 6.** BMP signaling in micropatterns and embryos correlates with embryonic and extraembryonic mesoderm fates. (**A**) Transverse cryosection of immunostained embryo in *Figure 6—figure supplement 1B*. Scale bar, 50 μm. (**B**) Quantification of pSMAD1/5/8 and BRACHYURY fluorescence intensity in E6.5 embryos. Cells within the epiblast, primitive streak (PS) and mesoderm were manually selected on confocal images of transverse cryosections in ImageJ as shown in right-hand panel. PS = BRACHYURY positive cells at embryo posterior. Mesoderm = cells positioned between VE and Epi. Quantification was carried out on three cryosections per embryo. N, number of cells. Data represents mean fluorescence intensity ± S.D. normalized to Hoechst fluorescence. (**C**) MIPs of immunostained colonies differentiated as in *Figure 2E*. Second panel depicts high magnification of colony edge. Scale bars, 100 μm. BRA, BRACHYURY; pS1/5/8, phosphorylated SMAD1/5/8. (**D**) Depiction of spatial patterning across multiple colonies.
*Figure 6 continued on next page*

*Figure 6 continued*

Each dot represents a single cell. (**E**) Quantification of voxel fluorescence intensity of pSMAD1/5/8 from colony center (0 μm) to edge (500 μm). Data represents average voxel intensity across multiple colonies. pSMAD1/5/8 colony numbers (n) in upper right corner. Data relative to maximum voxel intensity across the time course for each marker.

DOI: https://doi.org/10.7554/eLife.32839.015

The following figure supplements are available for figure 6:

**Figure supplement 1.** BMP signaling is active in the posterior primitive streak, embryonic and extraembryonic mesoderm.

DOI: https://doi.org/10.7554/eLife.32839.016

**Figure supplement 2.** Bypassing the WNT receptor alters spatial patterning.

DOI: https://doi.org/10.7554/eLife.32839.017

and SOX17 were expressed at the outer colony edge indicating that mesoderm differentiation was unaffected (*Figure 6—figure supplement 2C,D*). However, the BRACHYURY expression territory was expanded throughout the center of the colony (*Figure 6—figure supplement 2E,F*), recapitulating the expansion of BRACHYURY expression in CHIR-cultured embryos (*Sumi et al., 2013*). These data suggest that the transmission of signals or activity of inhibitors through receptors is key for setting up distinct cell fate domains within the flat-disc micropatterned colonies.

## The absence of BMP allows DE and AxM specification

BMP, WNT, ACTIVIN and FGF directed micropattern EpiLC differentiation towards posterior embryonic fates (posterior Epi, PS, embryonic and extraembryonic mesoderm), but not cell fates arising from the anterior PS (DE and AxM). Since the anterior PS is devoid of BMP signaling activity, we reasoned that removing BMP would replicate this signaling niche and create an environment permissive to specify anterior, but not posterior fates. EpiLCs were plated onto micropatterns and differentiated for 72 hr with FGF2, ACTIVIN A, BMP4 and WNT3A (referred to as +BMP), FGF2, ACTIVIN A and WNT3A (referred to as -BMP), or FGF2, ACTIVIN A and WNT3A with a small molecule inhibitor of BMP signaling, DMH1 (*Ao et al., 2012*) (referred to as BMPi) (*Figure 7A*). In +BMP conditions, nuclear pSMAD1/5/8 was observed in cells at the perimeter of colonies alongside CDX2 and SOX17, followed by a region of BRACHYURY expression and a central region of cells expressing SOX2 and low levels of FOXA2 (*Figure 7B–E*). In medium conditions lacking BMP (-BMP), the absence of BMP signaling activity was confirmed by lack of nuclear pSMAD1/5/8 (*Figure 7B,E*). Under these conditions, the domain of extraembryonic mesoderm, marked by CDX2, was lost (*Figure 7C,E*). Instead we observed elevated SOX17 in outer cells, which was now robustly coexpressed with FOXA2 (*Figure 7D–F*) representing DE (*Figure 3—figure supplement 1E*). We also observed a separate population of outer cells that coexpressed FOXA2 and BRACHYURY (*Figure 7F,G*), likely representing cells within the anterior PS, node or AxM (*Figure 3A*, yellow arrowheads,*Figure 3—figure supplement 1F*).

To further investigate the anterior cell fates formed in the absence of BMP, we differentiated $Gsc^{GFP/+}$; $Hhex^{RedStar/+}$ ESCs for 72 hr in the presence of FGF, ACTIVIN and WNT. While in the presence of BMP4, $Hhex^{RedStar}$ but not $Gsc^{GFP}$ was expressed (*Figure 3—figure supplement 2B*), in the absence of BMP, we observed FOXA2, BRACHYURY and $Gsc^{GFP}$ expression at the edge of micropatterned colonies from 24 hr of differentiation, followed by $Hhex^{RedStar}$ expression at 48 hr (*Figure 7—figure supplement 1A–C*). The number of FOXA2, BRACHYURY, $Gsc^{GFP}$ and $Hhex^{RedStar}$-expressing cells increased over time. The majority of $Hhex^{RedStar}$-positive cells coexpressed $Gsc^{GFP}$ and FOXA2 (*Figure 7—figure supplement 1D,E*) a signature of both DE and AxM. However, as we observed little overlap between $Gsc^{GFP}$, $Hhex^{RedStar}$ and the AxM marker BRACHYURY within individual cells (*Figure 7—figure supplement 1F,G*), $Gsc^{GFP}$/$Hhex^{RedStar}$/FOXA2 coexpression likely represented DE. The FOXA2/BRACHYURY-coexpressing cells observed in the absence of BMP (*Figure 7F,G*) may correspond to a subpopulation of anterior PS cells (*Figure 3A*) or alternatively AxM cells that have downregulated *Gsc* and *Hhex*. Global transcriptional analysis may be required to resolve these possibilities. We also frequently observed cells that coexpressed $Gsc^{GFP}$ and FOXA2 but not $Hhex^{RedStar}$ (*Figure 7—figure supplement 1D*), which may represent a BRACHYURY negative anterior PS-like state (*Figure 3A*) (*Burtscher et al., 2012*).

In conditions lacking BMP signaling activity, SOX2 levels were elevated relative to those in the presence of BMP (*Figure 7B–E*). This suggested that central cells represent a more anterior Epi

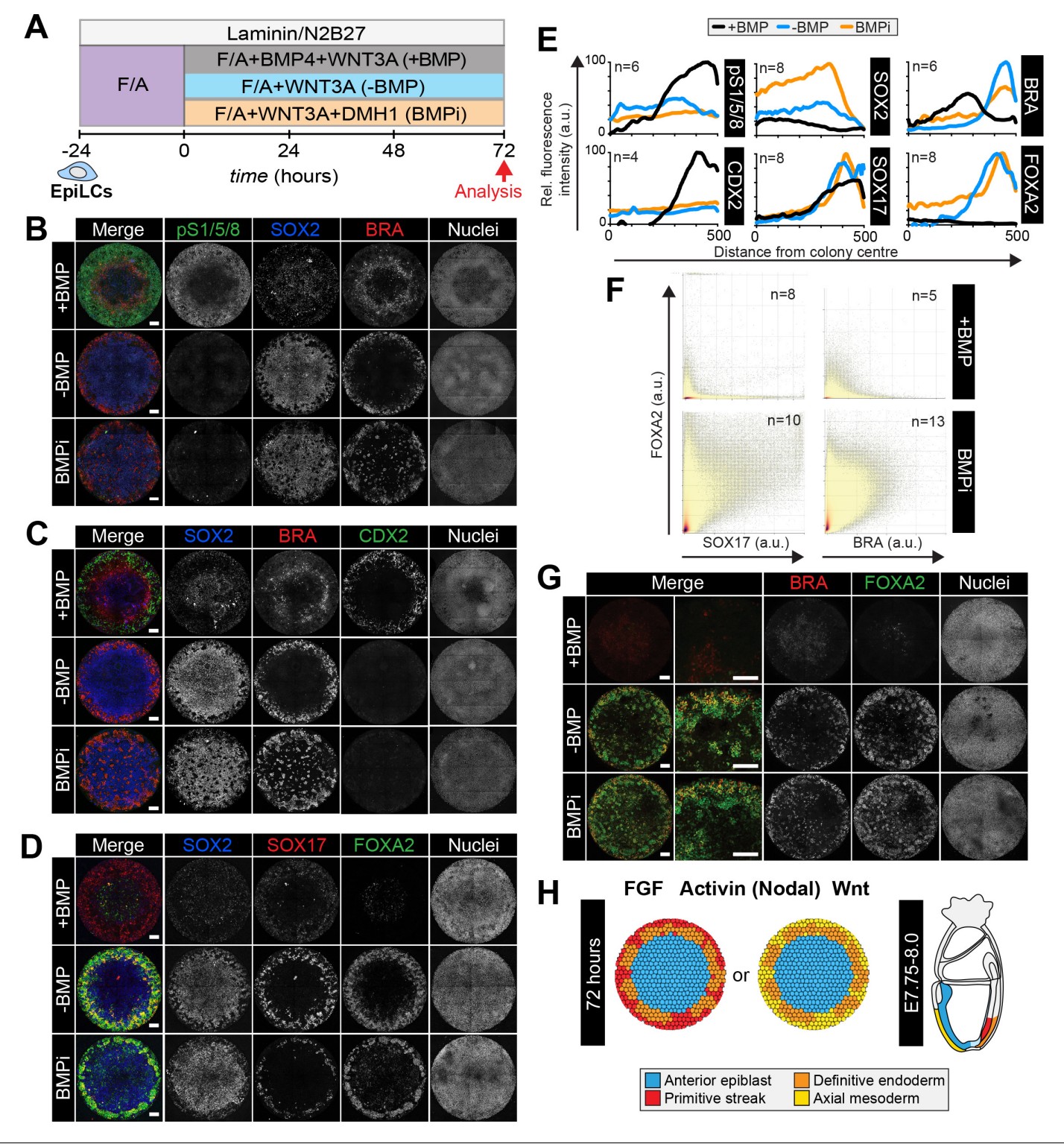

**Figure 7.** Anterior primitive streak fates are specified in the absence of BMP. (**A**) EpiLCs generated as in *Figure 1C* were plated overnight onto Laminin-coated micropatterns (−24 hr) in N2B27 medium with F/A. Various conditions were used for further differentiation - F/A, BMP4, WNT3A (+BMP), F/A, WNT3A (-BMP) or F/A, WNT3A with DMH1 BMP signaling inhibitor (BMPi). Colonies were analyzed after 72 hr differentiation. (**B–D, G**). MIPs of immunostained 72 hr colonies. Scale bars, 100 μm. (**E**) Quantification of immunostaining. Voxel fluorescence intensity was measured from colony center (0) to edge (500). Data represents average voxel intensity across multiple colonies relative to maximum voxel intensity for each marker. (**F**) Quantification of marker coexpression by voxel. Each dot indicates fluorescence intensity of a single voxel. Color represents voxel density within the

*Figure 7 continued on next page*

*Figure 7 continued*

plot. Numbers within quadrants show % of voxels within the gate. N, number of colonies. (**H**) Schematic diagram summarizing the cell fates observed after 72 hr in vitro differentiation under conditions described in A and corresponding in vivo cell types at E7.75-E8.0. The outer domain of the micropattern colony comprises cells that coexpress SOX17 and FOXA2, representing definitive endoderm and cells that coexpress BRACHYURY and FOXA2, representing anterior primitive streak or axial mesoderm cells.

DOI: https://doi.org/10.7554/eLife.32839.018

The following figure supplements are available for figure 7:

**Figure supplement 1.** Anterior primitive streak and definitive endoderm populations are formed in the absence of BMP.

DOI: https://doi.org/10.7554/eLife.32839.019

**Figure supplement 2.** An anterior epiblast/neurectoderm population is formed in the absence of BMP.

DOI: https://doi.org/10.7554/eLife.32839.020

state. To determine whether this was the case, we utilized a *Sox1$^{GFP}$* fluorescent reporter ESC line (*Ying et al., 2003*). *Sox1$^{GFP}$* marks early neurectoderm specification from cells within the anterior Epi (*Ying et al., 2003*). We differentiated *Sox1$^{GFP}$* ESCs as described in *Figure 7A*, either in the presence or absence of BMP. As with other cell lines analyzed, in the presence of BMP cells within the outer domain of *Sox1$^{GFP}$* EpiLC micropatterned colonies expressed CDX2 and, in the absence of BMP, they expressed FOXA2 (*Figure 7—figure supplement 2A,C*). While *Sox1$^{GFP}$* was largely absent from micropatterned colonies in the presence of BMP, consistent with the colony center representing posterior Epi, in the absence of BMP *Sox1$^{GFP}$* was expressed at high levels throughout the colony center (*Figure 7—figure supplement 2B,C*). Furthermore, in the absence of BMP, OTX2 levels were also elevated, with the highest expression observed at the colony periphery within the domain corresponding to DE and AxM fates (*Figure 7—figure supplement 2B,C*). This agrees with the later embryonic expression of OTX2 within the DE (*Figure 7—figure supplement 2D*). Hence, removing BMP from the (FGF, ACTIVIN and WNT) growth factor cocktail promoted differentiation towards anterior Epi, DE and anterior PS and/or AxM fates (*Figure 7H*, *Table 1*).

## Epiblast stem cells form definitive endoderm in the presence and absence of BMP

EpiSCs, maintained under standard FGF and ACTIVIN (F/A) culture conditions (*Tesar et al., 2007*; *Brons et al., 2007*), correlate to later embryonic stages than EpiLCs do (*Hayashi et al., 2011*). While EpiLCs represent the pre-gastrulation Epi, EpiSCs are similar to the Epi during gastrulation and express markers associated with the anterior PS (*Kojima et al., 2014b*). We therefore asked whether EpiSCs demonstrated a distinct differentiation capacity from EpiLCs in the context of the micropattern system.

EpiSC9 cells (*Najm et al., 2011*) were cultured in defined medium with 12 ng/ml FGF2 and 20 ng/ml ACTIVIN A. EpiSCs were plated onto the micropatterns as described for EpiLCs and differentiated in the same manner - for 72 hr in the presence or absence of BMP (*Figure 8A*). In the presence of BMP, differentiated EpiSC colonies showed an elevated expression of lineage-associated markers, including BRACHYURY, GATA6, FOXA2 and SOX17, at the colony periphery but lacked obvious spatial organization within more central regions (*Figure 8B,C*). GATA6 and SOX17/FOXA2 expression represented a DE fate while BRACHYURY was expressed at the outer colony edge in the same domain as FOXA2 corresponding to anterior PS or AxM cell types. SOX2-expressing cells were also present, likely representing an Epi-like state (*Figure 8B*). Under these conditions, EpiSCs generated few CDX2-positive cells indicating a significant reduction in the formation of extraembryonic mesoderm (*Figure 8B*). In the absence of BMP, GATA6, FOXA2 and SOX17 were expressed more uniformly throughout the colonies (*Figure 8B,C*). Hence, unlike EpiLCs, EpiSCs specified anterior cell fates both in the presence and absence of BMP.

We then utilized published microarray data from Hayashi et al (*Hayashi et al., 2011*), comparing the pre-gastrulation E5.75 Epi, EpiLCs and EpiSCs, to ask what may underlie this difference in the micropattern differentiation of EpiLCs and EpiSCs. As previously described (*Hayashi et al., 2011*; *Kojima et al., 2014b*) the E5.75 Epi, EpiLCs and EpiSCs all express high levels of the pluripotency marker *Pou5f1*, but EpiSCs also express high levels of the anterior markers *Foxa2* and *Sox17* (*Figure 8D*). Furthermore, EpiSCs show a marked increase in the expression of the BMP pathway inhibitor *Chordin*, the Wnt pathway inhibitor *Dkk1*, and the Nodal pathway inhibitor *Lefty2*

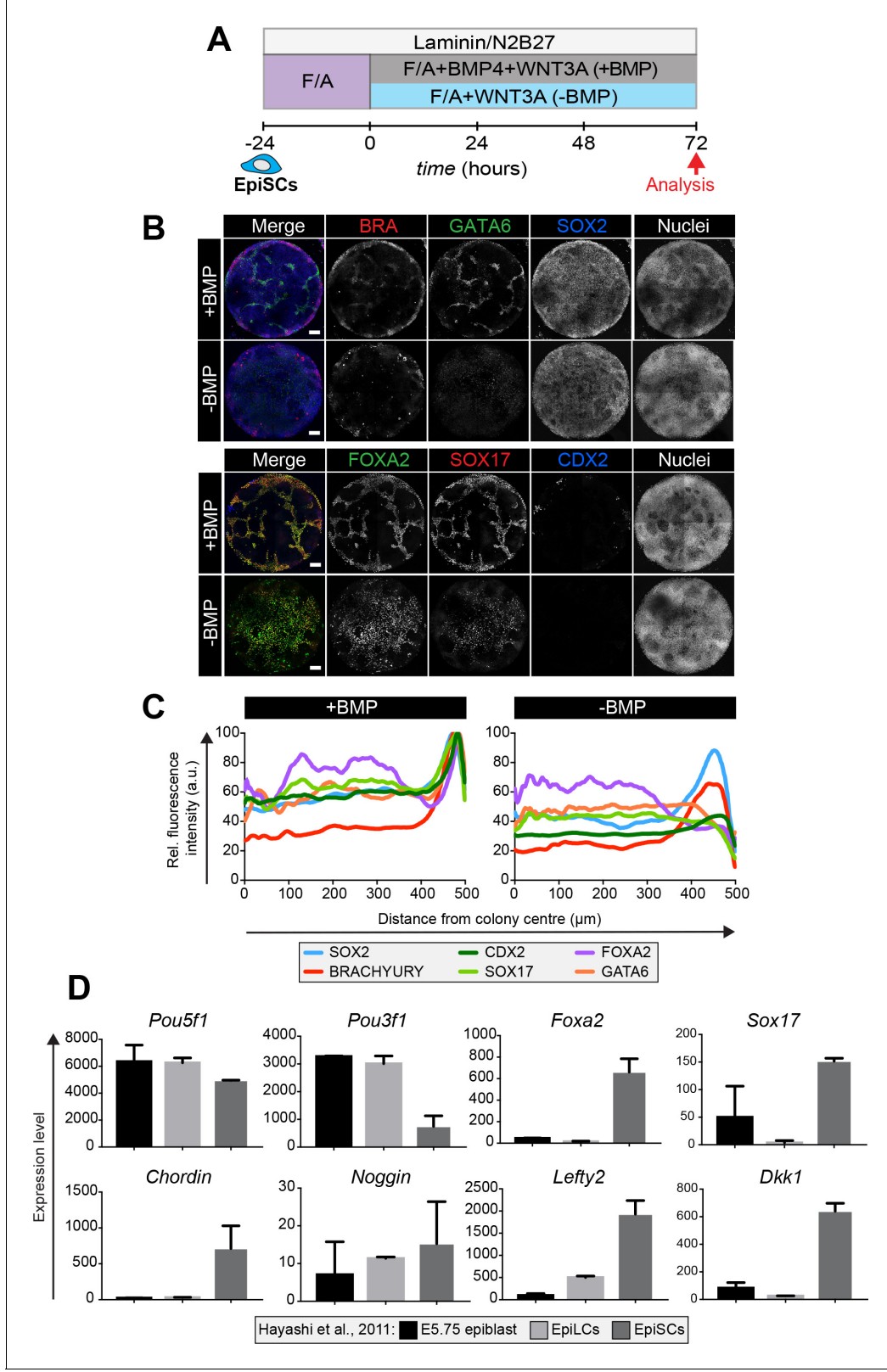

**Figure 8.** Epiblast stem cells undergo definitive endoderm differentiation in the presence or absence of BMP. (**A**) Epiblast stem cells (EpiSCs) of the EpiSC9 line (*Najm et al., 2011*) were cultured in the presence of 12 ng/ml FGF2 and 20 ng/ml ACTIVIN A (F/A) on fibronectin. EpiSCs were then plated overnight onto Laminin-coated micropatterns (−24 hr) in N2B27 medium with F/A. Various conditions were used for further differentiation - F/A, BMP4, WNT3A (+BMP) or F/A, WNT3A (-BMP). Colonies were analyzed after 72 hr differentiation. (**B**) MIPs of immunostained 72 hr colonies. Scale bars, 100

*Figure 8 continued on next page*

*Figure 8 continued*

μm. (**C**) Quantification of immunostaining. Voxel fluorescence intensity was measured from colony center (0) to edge (500). Data represents average voxel intensity across multiple colonies (n = 10/condition) and is shown relative to maximum voxel intensity for each marker across both conditions. (**D**) Graphs showing the expression level of a number of genes from the published microarray dataset of Hayashi et al. from E5.75 in vivo epiblast, EpiLCs and EpiSCs (*Hayashi et al., 2011*). Data shown is from amplified RNA samples and represents the mean ± S.D for two independent replicates.
DOI: https://doi.org/10.7554/eLife.32839.021

(*Figure 8D*) that, in this context, may render EpiSCs unresponsive to the BMP posteriorization signal.

## Discussion

We have developed a robust, quantitative and scalable micropattern protocol promoting the organized differentiation of mouse EpiLCs, the in vitro counterpart of the pre-gastrulation pluripotent Epi of the embryo (*Hayashi et al., 2011*). In response to FGF, ACTIVIN (NODAL), BMP and WNT, the critical gastrulation-inducing signals acting in the mouse embryo (*Arnold and Robertson, 2009*), EpiLCs grown on circular micropatterns underwent reproducible spatially coordinated cell fate specification comparable to in vivo gastrulation. Detailed marker analysis of gastrulating mouse embryos (which allow the mapping not only of marker expression but also of cell position) and micropatterns allowed us to link the in vitro differentiation to in vivo developmental time and space. In the absence of the spatial and temporal information of the embryo, we defined a cohort of 15 markers (SOX2, POU5F1, NANOG, SOX1, OTX2, BRACHYURY, CDX2, GATA6, SOX17, FOXA2, FOXF1, CDH1, CDH2, SNAIL, pSMAD1/5/8) that allowed us to distinguish between cell fates such as anterior versus posterior Epi, or extraembryonic mesoderm versus trophectoderm and DE, as these cell types express many common factors. This emphasizes the necessity of expression signatures, rather than individual markers, to accurately assign cell fates in vitro.

During 72 hr of differentiation, micropatterned colonies advanced from an E5.5 pluripotent Epi-like state to comprising an array of populations present in the embryo just prior to E7.75 (*Figure 9A*). Hence, under these culture conditions, in vitro cellular differentiation was slower than in vivo development. Conceivably, further manipulation of the timing, levels and combination of signaling factors provided to EpiLCs, as well as the extracellular matrix composition and stiffness of the substrate on which cells are maintained to more closely mimic that of the embryo, could alter the rate of differentiation and support the specification of cell fates emerging at later gastrulation stages.

At 72 hr, micropatterned colonies could be divided into three spatially distinct domains (central, mid and outer) (*Figure 9A*). Cells within the colony center showed minimal BMP signaling and expressed posterior Epi markers. PS markers were initially expressed at the periphery, but over time were observed more centrally. This was accompanied by an EMT and the emergence of outer mesenchymal cells, plausibly emanating from the PS-like region. The outer domain displayed elevated BMP activity and contained multiple populations including allantois and yolk sac extraembryonic mesoderm and early embryonic mesoderm.

In contrast to most gastrulating viviparous mammalian embryos, which exhibit a flat-disc geometry, rodents including the mouse are cup-shaped. A conceptual flattening of the cell fate arrangement within the mouse embryo (*Behringer et al., 2000*), could not fully recapitulate the organization of cell types observed within the flat-disc micropatterns. Therefore the most evident correspondence between embryonic and micropattern cell fates was signaling history. However, while all cells within the outer micropattern domain experienced high levels of BMP signaling, both embryonic and extraembryonic mesoderm fates were specified. It is therefore unclear whether additional morphogens distinguish embryonic and extraembryonic mesoderm, or if factors such as three dimensional growth, migration and extracellular matrix composition or substratum stiffness dictate fate. Extension of the micropattern system to different geometries, morphogens and inhibitors should resolve these questions.

Spatial organization of cell identities within the micropatterns emerged even though signals were provided uniformly. Thus, epithelial cell cultures can self-organize and the signaling history of a cell depends on its local environment, as well as the external medium. When WNT3 was replaced with

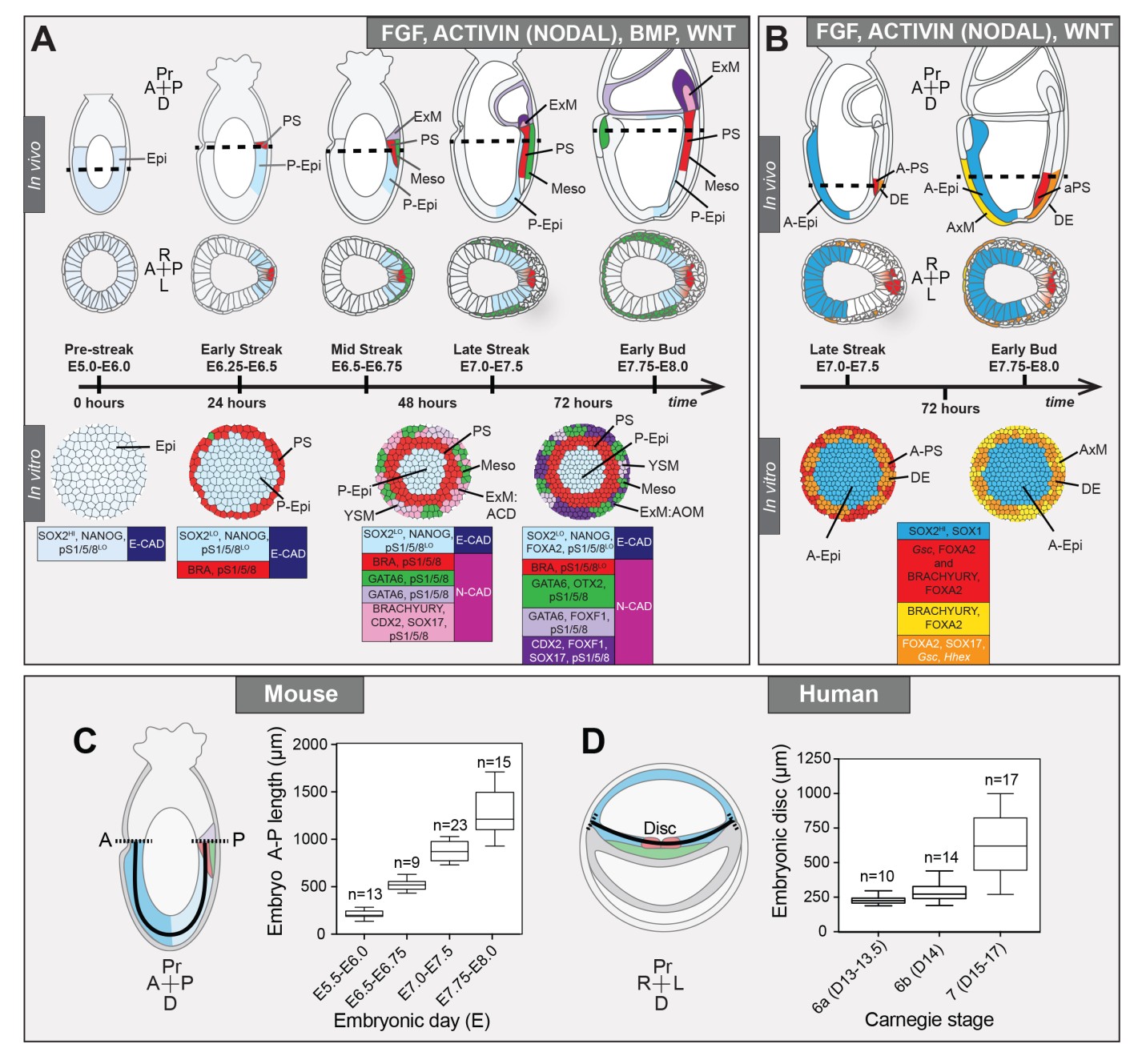

**Figure 9.** Micropattern differentiation of mouse pluripotent stem cells recapitulates cell fate specification in the posterior or anterior primitive streak. (A) Summary of embryo gastrulation (upper) and correlation with in vitro micropattern differentiation (lower). With FGF2, ACTIVIN A, BMP4 and WNT3A, mouse PSC differentiation recapitulated differentiation in the proximal posterior of the gastrulating embryo. Epi-like cells (EpiLCs) correlated to the embryonic day (E) 5.5–6.0 pre-streak epiblast (Epi). After 24 hr, cells in the colony center adopted a posterior Epi (P-Epi) identity and a primitive streak (PS)-like population arose at the colony edge, as E6.25-E6.5. After 48 hr, clusters of cell populations emerged at the outer colony edge correlating to embryonic Mesoderm 1 (Meso), and extraembryonic mesoderm (ExM) allantois core domain (ACD) arising at E6.75-E7.0. After 72 hr, cells in the colony center represented the distal P-Epi of E7.0-E7.25 embryos. Meso, ExM and PS populations were maintained. However, ACD cells were replaced by cells with an allantois outer mesenchyme (AOM) identity. Cells were highly confluent and could not be maintained under these conditions after 72 hr. LO, low expression. Dashed lines mark transverse plane shown below. (B) Summary of correlation between in vitro micropattern differentiation with FGF, ACTIVIN and WNT and in vivo gastrulating embryos. Under these conditions, mouse PSCs recapitulated differentiation of distal posterior (left panels) or distal posterior and anterior of embryo. After 72 hr, the central population expressed elevated levels of SOX2 compared to BMP4 conditions, likely representing anterior Epi (A-Epi). Cells coexpressed FOXA2, SOX17, *Gsc* and *Hhex* representing definitive endoderm (DE), FOXA2 and BRACHYURY representing anterior PS (A–PS) or AxM and or FOXA2 and *Gsc* likely representing A-PS. HI, high expression. A, anterior; P,

*Figure 9 continued on next page*

*Figure 9 continued*

posterior; Pr, proximal; D, distal. Color-coded legends highlight key markers of different cell states at each time point. (**C**) Box plots showing Epi length along the anterior-posterior (**A–P**) axis at pre- (E5.5-E6.0) early (E6.5-E6.75), mid- (E7.0-E7.5) and late (E7.75-E8.0) streak stages of mouse embryonic development. The A-P length was measured on sagittal confocal optical sections through the middle of the embryo with ImageJ software, as depicted in the schematic diagram. N, number of embryos. (**D**) Box plots showing human embryonic disc measurements compiled from human embryo data collections. Abnormal embryo data were excluded. D = embryonic day. Carnegie stage 6a, pre-streak; 6b, early streak; 7, early-mid gastrulation.

DOI: https://doi.org/10.7554/eLife.32839.022

CHIR (*Wray et al., 2011*) a small molecule that activates the WNT signaling pathway intracellularly, bypassing the receptors and secreted inhibitors acting at the cell surface, the PS region expanded into the colony center. Hence, as with human micropattern differentiation (*Warmflash et al., 2014*), endogenously produced inhibitors likely exclude signals from the colony center to define the inner domains. The identity of these inhibitors represents an open question to be elucidated in future studies.

In vivo, localized signaling from the extraembryonic tissues, notably the anterior visceral endoderm (AVE), induces molecular asymmetries within the bilaterally symmetrical Epi leading to anterior-posterior axis establishment (*Stower and Srinivas, 2014*). As the described micropattern system does not contain extraembryonic VE or ExE cells, and the disc-shaped colonies are also morphologically symmetrical, there is no apparent chemical or physical source of symmetry-breaking. Therefore, micropattern-differentiated PSCs might generate radially symmetric cell fate domains and be reminiscent of mutant embryos with defects in AVE specification or positioning (*Nowotschin et al., 2013*; *Migeotte et al., 2010*; *Kimura-Yoshida et al., 2005*; *Ding et al., 1998*), that lack the endogenous source of symmetry breaking. Interestingly, a mouse PSC-based three-dimensional differentiation system involving embryoid body-like aggregates, demonstrated asymmetric lineage marker expression in the absence of extraembryonic cell types (*van den Brink et al., 2014*). However, as these structures are not geometrically uniform, the polarized marker expression likely stems from initial morphological asymmetries and there is still no evidence of spontaneous symmetry breaking within geometrically uniform structures.

The cohort of signaling factors and secreted inhibitors expressed by adjacent tissues within the embryo make development robust yet difficult to quantify. For example, the extraembryonic VE is a source of inhibitors including CERBERUS and LEFTY1 on the anterior (*Stower and Srinivas, 2014*), and WNT3 on the posterior (*Rivera-Pérez and Magnuson, 2005*) side of the embryo, whereas the Epi and its derivatives express WNT3, LEFTY2 and DKK1 (*Peng et al., 2016*; *Meno et al., 1999*). As our in vitro system patterns in the absence of extraembryonic cell types, it allows us to decipher Epi-intrinsic patterning mechanisms.

The micropattern system can be used to extend findings in animal models to a defined, serum-free environment where signaling modulation can be unambiguously interpreted to reveal how timing and levels of signaling influence cell fate. As a first step in this direction, we analyzed the effect of manipulating the BMP pathway. Embryos with disrupted BMP signaling do not form a morphological PS and predominantly arrest at early gastrulation (*Gu et al., 1999*; *Mishina et al., 1999*; *Mishina et al., 1995*; *Winnier et al., 1995*), obscuring the assessment of a role for BMP in later mesoderm and endoderm specification. When we applied FGF, ACTIVIN and WNT alone (in the absence of BMP) to micropatterned colonies, anterior rather than posterior cell fates were specified (*Figure 7B*). These data revealed that BMP is not significantly induced by WNT and its absence does not perturb anterior cell fate specification. In the future, the micropattern assay could be used as a robust, efficient and scalable way to survey signaling conditions and systematically screen interactions between individual genes and pathways.

While a spectrum of mouse PSC states have been captured in vitro (*Morgani et al., 2017*), their comparative functional capacities and relation to the embryo is largely unknown. The micropattern system represents a quantifiable means to test the differentiation potential of PSC states and cell lines under defined conditions. Here we observed that EpiLCs patterned either posterior or anterior cell fates in the presence or absence of BMP respectively. Conversely, although EpiSCs can contribute to all germ layers in chimaera assays (*Huang et al., 2012*), within the micropattern system they predominantly generated DE cells and exhibited minimal self-organization. The limited capacity of EpiSCs to pattern in isolation may stem from their elevated expression of signaling inhibitors

(*Hayashi et al., 2011*). Alternatively, exogenous FGF, ACTIVIN, BMP and WNT may not be sufficient to induce the expression of secondary factors required for patterning and posterior cell fate specification of EpiSCs. Further manipulation of the micropattern differentiation conditions may give novel insights into the unique requirements of these different pluripotent cell states (*Figure 1A*) for organized cell fate specification.

Due to a paucity of data on gastrulating human embryos, cell fates arising during hESC micropattern differentiation can only been predicted (*Warmflash et al., 2014*; *Etoc et al., 2016*). The mouse micropattern differentiation provides the essential missing link between in vitro gastrulation models in mouse and human, and in vivo mouse development. The identification of an extraembryonic mesoderm population within mouse, but not human micropatterns prompts an analysis of the human system with equivalent marker combinations under comparable serum-free medium conditions containing both BMP4 and WNT3A to determine whether populations such as extraembryonic mesoderm can be generated, or whether in human, extraembryonic mesoderm, as has been shown for the amnion (*Dobreva et al., 2010*), does not arise from the Epi at gastrulation. Human and mouse embryos are of a similar size (*Figure 8C,D*), their corresponding in vitro PSCs undergo micropattern differentiation within equivalent diameter colonies and specify cell fates as a function of distance from the colony edge, suggesting that these species use common mechanisms to regulate cell fate specification and tissue patterning. The further correlation of mouse and human in vitro micropattern data, in the context of different pluripotent starting states, and corroborated with in vivo data from mouse embryos should yield insights into the conserved and divergent mechanisms regulating fundamental aspects of early mammalian development.

# Materials and methods

## Key resources table

| Reagent type (species) or resource | Designation | Source or reference | Identifiers | Additional information |
|---|---|---|---|---|
| Strain, strain background (*Mus musculus*) | Crl:CD1 (ICR) | | RRID:IMSR_CRL:22 | CD1 *Mus musculus* wild-type outbred mouse |
| Cell line (*Mus musculus*) | ES-E14 | (*Hooper et al., 1987*) | RRID:CVCL_C320 | Embryonic stem cell line: *Mus musculus* |
| Cell line (*Mus musculus*) | ES-R1 | (*Nagy et al., 1993*) | RRID:CVCL_2167 | Embryonic stem cell line: *Mus musculus* |
| Cell line (*Mus musculus*) | $Sox17^{GFP/+}$ | (*Kim et al., 2007*) | | Embryonic stem cell line: *Mus musculus* |
| Cell line (*Mus musculus*) | $T^{GFP/+}$ | (*Abe and Naski, 2004*) | | Embryonic stem cell line: *Mus musculus* |
| Cell line (*Mus musculus*) | 46 C cell line ($Sox1^{GFP}$) | (*Ying et al., 2003*) | RRID:CVCL_Y482 | Embryonic stem cell line: *Mus musculus* |
| Cell line (*Mus musculus*) | $Gsc^{GFP/+}$; $Hhex^{RedStar/+}$ | (*Villegas et al., 2013*) | | Embryonic stem cell line: *Mus musculus* |
| Cell line (*Mus musculus*) | EpiSC9 | (*Najm et al., 2011*) | | Epiblast stem cell line: *Mus musculus* |
| Antibody | anti-BRACHYURY | R and D Systems | Cat# AF2085, RRID:AB_2200235 | 1:200 |
| Antibody | anti-CDH1 | Sigma-Aldrich | Cat# U3254, RRID:AB_477600 | 1:500 |
| Antibody | anti-CDH2 | Santa Cruz Biotechnology | Cat# sc-7939, RRID:AB_647794 | 1:300 |
| Antibody | anti-CDX2 | BioGenex | Cat# AM392, RRID:AB_2650531 | 1:200 |
| Antibody | anti-DsRed | Clontech Laboratories, Inc. | Cat# 632496, RRID:AB_10013483 | 1:500 |
| Antibody | anti-FOXA2 | Abcam | Cat# ab108422, RRID:AB_11157157 | 1:500 |

*Continued on next page*

Continued

| Reagent type (species) or resource | Designation | Source or reference | Identifiers | Additional information |
|---|---|---|---|---|
| Antibody | anti-FOXF1 | R and D Systems | Cat# AF4798, RRID:AB_2105588 | 1:200 |
| Antibody | anti-GATA6 | R and D Systems | Cat# AF1700, RRID:AB_2108901 | 1:100 |
| Antibody | anti-GATA6 | Cell Signaling Technology | Cat# 5851, RRID:AB_10705521 | 1:500 |
| Antibody | anti-GFP | Aves Labs | Cat# GFP-1020, RRID:AB_10000240 | 1:500 |
| antibody | anti-KLF4 | R and D Systems | Cat# AF3158, RRID:AB_2130245 | 1:200 |
| Antibody | anti-NANOG | Thermo Fisher Scientific | Cat# 14-5761-80, RRID:AB_763613 | 1:200 |
| Antibody | anti-NANOG | Cosmo Bio Co | Cat# REC-RCAB0002PF, RRID:AB_567471 | 1:500 |
| Antibody | anti-OTX2 | R and D Systems | Cat# AF1979, RRID:AB_2157172 | 1:500 |
| Antibody | anti-POU3F1 | Millipore Sigma | MABN738 | 1:100 |
| Antibody | anti-POU5F1 | Santa Cruz Biotechnology | Cat# sc-5279, RRID:AB_628051 | 1:100 |
| Antibody | anti-pSMAD1/5/8 | a gift from Dr. Ed Laufer, Columbia University, New York, NY | N/A | 1:200 |
| Antibody | anti-SNAIL | R and D Systems | Cat# AF3639, RRID:AB_2191738 | 1:100 |
| Antibody | anti-SOX2 | Thermo Fisher Scientific | Cat# 14-9811-82, RRID:AB_11219471 | 1:200 |
| Antibody | anti-SOX17 | R and D Systems | Cat# AF1924, RRID:AB_355060 | 1:200 |
| Software, algorithm | Ilastik | http://ilastik.org/ | RRID:SCR_015246 | 3-D Nuclear mask generation |

## Gene and gene product nomenclature

Genes and gene products are referred to using guidelines set by the International Committee on Standardized Genetic Nomenclature for Mice - gene symbols are italicized with only the first letter upper case while proteins are all upper case and no italics (http://www.informatics.jax.org/mgihome/nomen/gene.shtml). Cytokines are referred to as proteins (all upper case) while the corresponding signaling pathways are referred to in lower case, non-italic.

## Cell culture

ESC lines used for this study include E14 (129/Ola background) (*Hooper et al., 1987*), R1 (129/Sv background) (*Nagy et al., 1993*), *Sox17$^{GFP/+}$* (R1, 129/Sv background) (*Kim et al., 2007*), *T$^{GFP/+}$* (E14.1, 129/Ola background, also known as GFP-Bry) (*Fehling et al., 2003*) and *Sox1$^{GFP}$* (E14Tg2a background, also known as 46C) (*Ying et al., 2003*), *Gsc$^{GFP/+}$*; *Hhex$^{RedStar/+}$* (E14Tg2a background) (*Villegas et al., 2013*). ESCs were routinely cultured on 0.1% gelatin coated tissue culture grade plates (Falcon, Tewksbury, MA) in serum and LIF medium as previously described (*Morgani et al., 2013*). Serum and LIF medium was comprised of Dulbecco's modified Eagle's medium (DMEM) (Gibco, Gaithersburg, MD) containing 0.1 mM non-essential amino-acids (NEAA), 2 mM glutamine and 1 mM sodium pyruvate, 100 U/ml Penicillin, 100 µg/ml Streptomycin (all from Life Technologies, Carlsbad, CA), 0.1 mM 2-mercaptoethanol (Sigma, St. Louis, MO), and 10% Fetal Calf Serum (FCS, F2442, Sigma) together with 1000 U/ml LIF. They were passaged every 2 days upon reaching approximately 80% confluence by washing with phosphate buffered saline (PBS) before adding 0.05% Trypsin (Life Technologies) for 3 min at 37°C and dissociating into a single cell suspension by pipetting. Trypsin activity was then neutralized with serum-containing medium. Cells were collected at 1300 rpm for 3 min and 1/5 of cells transferred to a new plate.

For this study, the EpiSC9 epiblast stem cell line was used (129SvEv x ICR background) (*Najm et al., 2011*). EpiSCs were cultured under standard conditions as previously described (*Brons et al., 2007*), in defined, serum-free N2B27 medium with 12 ng/ml FGF2 and 20 ng/ml ACTIVIN A. EpiSCs were passaged upon reaching approximately 80% confluence by washing with PBS then replacing with Accutase (Sigma) and scraping cells from the plate. Cells were pipetted gently to avoid single cell dissociation. Cells were collected at 1300 rpm for 3 min and 1/5 of cells transferred to a new plate. ESCs and EpiSCs were maintained at 37°C at 5% $CO_2$ and 90% humidity.

## EpiLC conversion

Prior to plating on micropatterns, ESCs were converted to a transient EpiLC state as previously described (*Hayashi et al., 2011*). First, 10 cm plates were coated with 16 µg/ml of Fibronectin (FC010, Millipore Sigma, Billerica, MA) for 1 hr at room temperature followed by two washes with PBS. ESCs were collected by trypsinization (see above), counted and $1.6 \times 10^6$ cells plated onto the Fibronectin-coated plates for 48 hr in EpiLC medium, N2B27 medium containing 20 ng/ml ACTIVIN A and 12 ng/ml FGF2 (Peprotech, Rocky Hills, NJ). Medium was changed daily.

## Micropattern differentiation

To coat micropatterned surfaces, a solution was prepared of 20 µg/ml Laminin (L20202, Sigma) in PBS without calcium and magnesium (PBS-/-). A 15 cm tissue culture plate was lined with Parafilm (Pechiney Plastic Packaging, Chicago, IL) and 700 µl drops were made onto the Parafilm surface. Micropatterned chips (Arena A, CYTOO, France) were washed once with PBS-/- and then inverted on top of the drops followed by incubation at 37°C for 2 hr. Micropatterns were then washed 5 times with 5 ml of PBS-/-. EpiLCs were collected by trypsinization and a single cell suspension generated. Cells were counted and $2 \times 10^6$ EpiLCs were evenly plated onto micropatterns within 6-well plates (Falcon) in EpiLC medium. Medium was supplemented with a small molecule inhibitor of Rho-associated kinase (ROCKi, 10 µM Y-27632, Tocris Bioscience, UK) for the first 2 hr after plating, to reduce apoptosis (*Ohgushi et al., 2010*; *Watanabe et al., 2007*). Plates were maintained in the tissue culture hood for 30 min after plating to allow time for cells to evenly adhere to the micropatterns before moving to the incubator. After 2 hr, medium containing ROCKi was exchanged for N2B27 medium containing 12 ng/ml FGF2, 20 ng/ml ACTIVIN A, 50 ng/ml BMP4 (Peprotech) and 200 ng/ml WNT3A (R and D, Minneapolis, MN). Cells were maintained for up to 72 hr in this state, after which time cells were highly confluent and cell death was observed. To determine the effect of BMP signaling on the differentiation, cells were differentiated as described above for 72 hr with FGF, ACTIVIN, BMP and WNT (+BMP) or with FGF2, ACTIVIN A and WNT3A without BMP4 (-BMP) or FGF, ACTIVIN and WNT with the addition of 2 µM DMH1 (Sigma) (BMPi).

## Immunostaining, imaging and quantification of cells

Prior to immunostaining, cells were either grown on micropatterns or in 1 µ-slide eight well IbiTreat plates (Ibidi, Germany). Cells were washed twice with PBS before being fixed with 4% paraformaldehyde (PFA) (Electron Microscopy Sciences, Hatfield, PA) at room temperature for 15 min. Cells were then washed a further two times with PBS followed by permeabilization with PBS containing 0.1% Triton-X (Sigma) (PBS-T) for 10 min at room temperature. Cells were then blocked in PBS-T with 1% bovine serum albumin (BSA, Sigma) and 3% donkey serum (Sigma) for 30 min at room temperature. Primary antibodies were added overnight at 4°C, diluted to the appropriate concentration in PBS-T with 1% BSA. Details of primary antibodies are supplied in Key Resources Table. The following day, cells were washed three times for 15 min with PBS followed by incubation with secondary antibodies (1:500, Alexa Fluors, Life Technologies, Dylight, Jackson ImmunoResearch) in PBS-T with 1% BSA for 2 hr at room temperature. Finally, cells were washed three times for 15 min with PBS with the final wash containing 5 µg/ml Hoechst (Life Technologies). Cells grown on micropatterns were then mounted onto glass slides (Fisher Scientific, Hampton, NH) with Fluoromount-G (Southern Biotech, Birmingham, AL). Cells were imaged using a LSM880 confocal (Zeiss). Brightfield-only images were acquired using a Zeiss Axio Vert.A1.

## Quantitative analysis of micropattern differentiation

For micropattern image analysis and quantification, tiled Z-stack images of individual colonies were collected using a LSM880 confocal microscope (Zeiss) at 512 × 512 format using a 20x objective. The background signal was subtracted using ImageJ software and each channel saved as a separate tiff file. Tiff files containing the Hoechst nuclear staining of each colony were classified into regions containing nuclei and those that did not using Ilastik (*Sommer, 2011*), an interactive image classification software. Using this information, a 3D probability mask was generated and analysis carried out using custom software written in Python. All analysis was carried out on entire Z-stacks of multiple colonies and an average of results across colonies displayed.

Segmentation of individual cells within images of colonies proved problematic due to the large number and high density of cells. For these reasons, manual correction of segmentation, as routinely used in smaller systems (*Saiz et al., 2016*), was not feasible. Therefore quantification of immunostaining fluorescence intensity across the radii of colonies as well as coexpression analysis was completed on a voxel basis to eliminate segmentation artifacts. To generate plots of radial immunostaining fluorescence intensity, each voxel within a colony was assigned a distance from the colony center. The fluorescence intensity for each marker was measured per voxel and then the average fluorescence intensity of voxels at a particular radial position (binned into discrete radial bands) was calculated for each colony. The average radial fluorescence intensity across multiple colonies was then calculated. To display the expression of multiple different markers across the radii of colonies on the same scale, the relative level of each marker was quantified by normalizing to the highest level of expression (shown as 100) either across a time-course or within an individual time-point. Spatial patterning across multiple colonies was also demonstrated by generating average colony images for individual markers where each segmented cell was represented as a dot whose color indicates its fluorescence in the specified channel.

Coexpression analysis was carried out on a voxel level, that is the fluorescence level of each marker within a single voxel was calculated and plotted. For genes that were not expressed, or only expressed at low levels, at the start of the differentiation, gates could be drawn based on the fluorescence at 0 hr and used to quantify the percentage of total voxels expressing a particular marker at later time points.

## Nuclear density measurements in micropatterns and embryos

The number of nuclei per 100 μm was quantified for 0 hr and 24 hr of micropattern differentiation utilizing the colony side view (z-axis) from confocal images acquired using a 40x objective at 0.5 μm interval steps. The number of nuclei was quantified across the entire width of the colony at 10 distinct positions and the average number of nuclei per 100 μm distance were calculated. For E5.5 embryos, the number of nuclei per 100 μm was quantified on sagittal confocal optical sections based on the number of nuclei within a sagittal optical section of the epiblast and the distance around the epiblast within the same section, manually measured using ImageJ software. For E6.5 the same was done using confocal images of transverse cryosections. Only cells within the epiblast were counted. Five embryos at E5.5 and five at E6.5 were analyzed in this manner.

Inter-nuclear distance was manually measured using ImageJ software. A line was drawn from the center of one nuclei to the center of the adjacent nuclei. For micropattern differentiation, 150 measurements were made per time point (0 hr and 24 hr). For in vivo data, five different embryos were measured at each time point (E5.5 and E6.5). At E5.5, 125 measurements were made and at E6.5, 189 measurements were made.

## Mice

All mice used in this study were of a wild-type CD1 background. Mice were maintained in accordance with the guidelines of the Memorial Sloan Kettering Cancer Center (MSKCC) Institutional Animal Care and Use Committee (IACUC). Mice were housed under a 12 hr light/dark cycle in a specific pathogen free room in the designated facilities of MSKCC. Natural matings of CD1 males and 4–6 weeks old virgin CD1 females were set up in the evening and mice checked for copulation plugs the next morning. The date of vaginal plug was considered as E0.5.

## Immunostaining and imaging of embryos

To analyze the expression of markers within post-implantation embryos, the uterus of pregnant mice was dissected and deciduae removed. Embryos were dissected from the deciduae and the parietal endoderm removed. Embryos were washed twice in PBS and fixed in 4% PFA for 30 min at room temperature. Embryos were permeabilized in PBS with 0.5% Triton-X for 30 min followed by blocking overnight in PBS-T with 5% horse serum (Sigma). Primary antibodies were added the following day, diluted in blocking buffer at the appropriate concentration (details can be found in Key Resources Table) and incubated overnight at 4°C. The next day, embryos were washed 3 times for 15 min in PBS-T and then blocked for a minimum of 2 hr. Embryos were then incubated with the secondary antibodies diluted in blocking buffer overnight at 4°C. Alexa Fluor (Thermo Fisher Scientific) secondary antibodies were diluted 1:500. The following day, embryos were washed 3 times for 15 min in PBS-T with the last wash containing 5 μg/ml Hoechst. Embryos were imaged in PBS-T in glass bottom dishes (MatTek, Ashland, MA) using an LSM880 confocal (Zeiss).

## Cryosectioning and quantitative embryo measurements

For cryosectioning, embryos were incubated in a 30% sucrose solution until they sank to the bottom of the vial. Embryos were then transferred to optimal cutting temperature compound (Tissue-Tek® OCT, Sakura Finetek, Torrance, CA) overnight. The following day, embryos were transferred to mounting molds (Fisher Scientific) containing OCT and appropriately oriented to give sagittal or transverse sections. Embryo-containing molds were carefully transferred to dry ice until frozen and then temporarily to −80°C until cryosectioning. Cryosections of 10 μm were cut using a Leica CM3050S and imaged using a confocal microscope as described above.

To quantify immunostaining within gastrulating mouse embryos, transverse cryosections were imaged by confocal microscopy. For quantification of the relative levels of SOX2 and NANOG within different regions of the Epi, the anterior and posterior regions were manually selected using ImageJ software and immunostaining fluorescence levels in arbitrary units. Five cryosections were quantified per embryo and the levels normalized to the fluorescence levels of the Hoechst nuclear stain. At E6.5, three embryos were quantified, while as E7.5, two embryos were quantified. For quantification of the levels of pSMAD1/5/8 within different cell types within the gastrulating mouse embryo, transverse cryosections through the PS of E6.5 embryos were selected. Individual cells within the Epi, PS and mesodermal wings were manually selected using ImageJ software and fluorescence levels in arbitrary units. Data was normalized to the fluorescence level of the Hoechst nuclear stain. The PS was defined as BRACHYURY-expressing cells within the posterior Epi while the mesodermal wings were identified as cells that had left the Epi epithelial layer and were migrating between the Epi and VE. Three cryosections were quantified per embryo and three embryos were analyzed.

The diameter of embryos at different developmental stages was measured on acquired images using ImageJ software. Measurements were made along the anterior-posterior axis of transverse cryosections of embryos. Multiple cryosections at the widest region of the embryo were utilized and multiple embryos per developmental stage.

## Acknowledgements

We thank Paul Tesar for EpiSC lines; Josh Brickman for *Gsc*$^{GFP/+}$; *Hhex*$^{RedStar/+}$ and Austin Smith for *Sox1*$^{GFP}$ mESCs; Kathryn Anderson, Ali Brivanlou and members of the Hadjantonakis and Brivanlou-Siggia labs for critical discussions and comments on the manuscript. SMM is supported by a Wellcome Trust Sir Henry Wellcome postdoctoral fellowship under the supervision of JN and AKH. Work in the Hadjantonakis lab was supported by grants from NYSTEM (C029568) and the NIH (R01DK084391 and P30CA008748). Work in the Siggia lab was supported by the NSF (PHY1502151) and NIH (R01HD080699).

# Additional information

## Funding

| Funder | Grant reference number | Author |
|---|---|---|
| Wellcome Trust | | Sophie M Morgani |
| Eunice Kennedy Shriver National Institute of Child Health and Human Development | R01HD080699 | Eric D Siggia |
| National Science Foundation | PHY1502151 | Eric D Siggia |
| National Institute of Diabetes and Digestive and Kidney Diseases | R01DK084391 | Anna-Katerina Hadjantonakis |
| National Cancer Institute | P30CA008748 | Anna-Katerina Hadjantonakis |
| NYSTEM | C029568 | Anna-Katerina Hadjantonakis |

The funders had no role in study design, data collection and interpretation, or the decision to submit the work for publication.

## Author contributions

Sophie M Morgani, Conceptualization, Resources, Data curation, Formal analysis, Funding acquisition, Validation, Investigation, Visualization, Methodology, Writing—original draft, Writing—review and editing; Jakob J Metzger, Software, Formal analysis, Writing—review and editing; Jennifer Nichols, Resources, Supervision, Funding acquisition, Writing—review and editing; Eric D Siggia, Conceptualization, Software, Supervision, Funding acquisition, Writing—original draft, Project administration, Writing—review and editing; Anna-Katerina Hadjantonakis, Conceptualization, Resources, Supervision, Funding acquisition, Visualization, Writing—original draft, Project administration, Writing—review and editing

## Author ORCIDs

Sophie M Morgani http://orcid.org/0000-0002-4290-1080
Eric D Siggia http://orcid.org/0000-0001-7482-1854
Anna-Katerina Hadjantonakis http://orcid.org/0000-0002-7580-5124

## Ethics

Animal experimentation: Animal experimentation: All mice used in this study were maintained in accordance with the guidelines of the Memorial Sloan Kettering Cancer Center (MSKCC) Institutional Animal Care and Use Committee (IACUC) under protocol number 03-12-017 (PI Hadjantonakis).

## Decision letter and Author response

Decision letter https://doi.org/10.7554/eLife.32839.028
Author response https://doi.org/10.7554/eLife.32839.029

# Additional files

## Supplementary files

• Supplementary file 1. Summary of expression data for a panel of factors used for cell fate assignments in this study. Where data was not generated in this study, appropriate literature references are provided – listed in bold lowercase letters, full references provided below the table. Where primary data is included in this study, the reference is given in the following format – (Main text figure (Fig._) – figure supplement (S_): *Supplementary file 2*). Cell fates are color-coded as in *Figure 3—figure supplement 1A* and in table legend. At each gastrulation stage, each cell type was classified as 1 = factor expressed, 0.5 = factor expressed heterogeneously or at low levels, 0 = not expressed (white), ?=expression unclear from immunostaining and exact localization is unclear/unknown from

published literature. FOXF1 is expressed in the embryonic mesoderm and also reported to be expressed within the PS at E7.0-E7.5 but PS expression is unclear. *Gsc* and *Hhex* are both expressed in the AxM although their expression has been reported to be transient in some studies and the exact timing of expression is not clear. To note, some cells of the PS coexpressed BRACHYURY, SOX2 and NANOG, likely a transition state between Epi and PS before SOX2 and NANOG are downregulated. It was unclear whether GATA6 was absent or expressed only at low levels in the ACD and early AOM. Certain markers were expressed in some but not all embryos at a particular stage – likely indicating the onset of expression, for example SOX17 within the ExM at E7.0-E7.5. **\*\*\*** At E7.0-E7.5 both DE and mesoderm cells arise from the anterior PS and are spatially intermixed hence it is difficult to identify DE cells based only on localization. At this time DE therefore refers to presumptive DE cells arising from the anterior PS based on known marker expression.
DOI: https://doi.org/10.7554/eLife.32839.023

• Supplementary file 2. Representative data of immunostained gastrulating mouse embryos utilized to identify marker signatures of distinct cell states. Supplemental file comprising representative images of embryo data used in this study to identify in vivo marker signatures for particular cell states. Embryos were collected at different stages of development throughout gastrulation, from embryonic day (E) 6.25-E8.5 and immunostained for trios of marker combinations. File contains images of wholemount embryos and cryosections acquired by confocal microscopy. The expression patterns of the markers determined using this data are summarized in *Supplementary file 1*.
DOI: https://doi.org/10.7554/eLife.32839.024

## Major datasets

The following previously published dataset was used:

| Author(s) | Year | Dataset title | Dataset URL | Database, license, and accessibility information |
|---|---|---|---|---|
| Hayashi K, Kurimoto K, Ohta H, Saitou M | 2011 | Reconstitution of the mouse germ-cell specification pathway in culture by pluripotent stem cells | https://www.ncbi.nlm.nih.gov/geo/query/acc.cgi?acc=GSE30056 | Publicly available at the NCBI Gene Expression Omnibus (accession no: GSE30056) |

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
