## [Decision Letter]

Thank you for submitting your article "Micropattern differentiation of mouse pluripotent stem cells recapitulates embryo regionalized fates and patterning" for consideration by *eLife*. Your article has been favorably evaluated by Fiona Watt (Senior Editor) and three reviewers, one of whom is a member of our Board of Reviewing Editors. The following individuals involved in review of your submission have agreed to reveal their identity: Ronald McKay (Reviewer #2); Valerie Wilson (Reviewer #3).

The reviewers have discussed the reviews with one another and the Reviewing Editor has drafted this decision to help you prepare a revised submission.

Summary:

In this study Morgani et al. describe a micropatterned mouse pluripotent stem cell culture system that models many aspects of early post-implantation embryo development. Beginning with Epiblast like cells (EpiLC), which approximate the stage of formative pluripotency, the authors show that the system can recapitulate primitive streak formation, as well as anterior mesendoderm and definitive endoderm patterning, based on expression of combinations of cell type specific markers within the cultures. The authors validate the system by showing that treatment with appropriate morphogens can bias in vitro patterning in a fashion consistent with the actions of BMP, WNT, FGF and ACTIVIN signaling in the embryo.

Essential revisions:

All three reviewers are enthusiastic about the value of this experimental system and agree that these findings will be of interest to the field. Reviewer 2 raises some questions concerning the interpretation of the data on EpiSC that should be addressed, as does reviewer 3. Please comment on these points.

"When grown on a specific substrate the EpiLCs generated a series of intermediate states found in the embryonic and extra-embryonic posterior regions of the gastrulating mouse embryo. ES and EpiSCs do not show the same the same responses to the growth conditions used here. This difference is interpreted as suggesting the EpiLC is the common pluripotent state shared by mouse and human embryos. While this manuscript reports important data, is it not possible that the in vitro conditions used here do not adequately support all the cell fates adopted by either mouse EpiSCs or human induced pluripotent stem (iPS) cells?

More specifically, the formation of anterior neural fates was not extensively explored in this analysis. The conclusion of this study that EpiLCs are biased towards posterior embryonic and extra-embryonic mesodermal fates suggests that EpiSCs may be biased towards more anterior fates. This raises the possibility that EpiSCs may express functions normally found in the anterior extra-embryonic visceral endoderm. This possibility was not definitively excluded by the analysis presented here that is based on a restricted number of antibodies and did not employ genome wide analytical tools limiting the number of cellular states that might be detected. "

"One of the conclusions is that EpiSCs are 'functionally restricted in developmental potential' (Results). However, EpiSCs can differentiate to a lot of cell types (see Huang et al., 2012): is there an argument that this micropattern culture has not fully revealed the spectrum of EpiSC differentiation?"

Reviewer 2 requests some clarification regarding the unique outcomes of micropatterned cultures described here compared to other previous results.

"Results: 'Micropatterned cultures do not contain extraembryonic cell types (namely primitive endoderm or trophectoderm derivatives), they instead formed radially symmetric cell fate'. However, other 3d culture systems (e.g. van den Brink et al., 2014) generate asymmetry without extraembryonic cell types. Can the authors comment on how these differences could arise?"

All reviewers note some concerns about the identification of specific cell types. Reviewers 1 and 2 suggest that citation of current literature might help support the identification of particular cell types. Reviewer 3 raises a number of specific queries concerning particular markers; like reviewer 2 this individual is concerned to know whether particular markers were assessed at specific stages. It might be helpful to present the information on the markers in a somewhat more organized form. Figure S8 could be revised (for some reason I could not find the legend to this figure) to indicate which markers were tested at what stage, to indicate what figures show expression in the micropatterned cultures as well as the embryos, to include callouts to those data which are not identified with a particular figure in the manuscript, to show what the levels in the heatmap mean, and to include references from the literature where applicable. Excel might be the appropriate format for presenting this information.

"Since there was no attempt to characterise the cultured cells further, either through molecular techniques (e.g. single cell gene expression analysis using QRT-PCR or RNA-seq) or through studies of their further differentiation into appropriate lineages, the validation of these markers is particularly important. It is not easy to assess the fidelity with which the in vitro system replicates cells in the embryo on this basis of a few markers; it is possible that global examination of gene expression patterns might not support the identification of cell types, or that the cells in vitro might not have the developmental potential implied by the labels the authors assign to them. These limitations are shared by almost all studies of in vitro patterning to date, and at some stage, the field will have to progress beyond this level. At least, the basis for claiming that 2 or 3 markers can uniquely identify the cell types of interest (as tabulated in Supplemental Figure 3 and Figure 8) should be clearly specified: is it the immunostaining in the manuscript, or can the unique identifiers be further supported by gene expression data from the literature. Supplementary Figure 8 is quite critical, linking the data to assignment of cellular identities. Perhaps the supplemental material could group together the embryo images with staining identifying particular cell types through various stages. I found it awkward to keep flipping back and forth to check this information."

See also these specific comments on markers for cell identification:

*Reviewer 1:*

1) Subsection “EpiLCs recapitulate differentiation and spatial organization of posterior cell fates”. Figure 2 what are the *Sox2* positive cells in the extraembryonic region?

2) Figure 2. What staining is shown in the top two panels of Figure 2?

3) Subsection “EpiLCs recapitulate differentiation and spatial organization of posterior cell fates”. Figure 2 – what are the Cdx2 positive cells and what is the basis for their identification?

4) Subsection “EpiLCs recapitulate differentiation and spatial organization of posterior cell fates”, fifth paragraph. Figure 3—figure supplement 1 – not clear that Otx2 is expressed in posterior epiblast from this image; indicate where is posterior.

5) Figure 3 – indicate in the legend what the little red dots do i.e. delineate region of interest.

6) High resolution high magnification images of the embryos should perhaps be compiled somewhere to support the use of the markers in question.

*Reviewer 2:*

Along similar lines it would make the study more influential if the contemporary literature in early mouse development was more precisely cited. For example, the differences in the expression patterns of Nanog and Brachyury in the gastrulating mouse embryo that are central to the current report have been previously reported raising the important possibility that Nanog may have distinct functions in the inner cell mass and the primitive streak.

*Reviewer 3:*

1) It would be good to re-check all the specifics in the rubric. Some queries that arose from using this as the overview to all the data, including:a) Was Nanog immunostaining done at all stages? It is listed as posterior epiblast throughout but I don't see the immunostains this refers to.

b) Sox1-GFP should be added to the rubric to identify anterior epiblast.

c) The rubric lists 'Figure 5' as a source of information – I couldn't find panel 5I.

d) Definitive endoderm at E7.5-8.0 should list Figure 3—figure supplement 1 as a source.

e) Was OTX2 stain done in E7.0-7.25 embryos?

2) Some of the conclusions may need qualifying. For example check 'cells that coexpressed GATA6 and OTX2 (Figure 3—figure supplement 1), corresponding to Mesoderm 1': GATA6 and OTX2 are also expressed in endoderm. Is the argument that this is not endoderm because the endoderm only later expresses OTX2? Timings of individual markers may be a bit different in vitro. However, focusing on the most likely cell type is correct; just try and avoid overstating the certainty of a conclusion.

---

## [Author Response]

Essential revisions:All three reviewers are enthusiastic about the value of this experimental system and agree that these findings will be of interest to the field. Reviewer 2 raises some questions concerning the interpretation of the data on EpiSC that should be addressed, as does reviewer 3. Please comment on these points."When grown on a specific substrate the EpiLCs generated a series of intermediate states found in the embryonic and extra-embryonic posterior regions of the gastrulating mouse embryo. ES and EpiSCs do not show the same the same responses to the growth conditions used here. This difference is interpreted as suggesting the EpiLC is the common pluripotent state shared by mouse and human embryos. While this manuscript reports important data, is it not possible that the in vitro conditions used here do not adequately support all the cell fates adopted by either mouse EpiSCs or human induced pluripotent stem (iPS) cells?

We thank the reviewer for this comment. It is true that the in vitro micropattern conditions may not fully recapitulate all of the factors contributing to cell fate specification and patterning in the embryo. Manipulation of the conditions utilized for micropattern differentiation may give further insight into this possibility and we have modified the Discussion to reflect this:

“The limited capacity of EpiSCs to pattern in isolation may stem from their elevated expression of signaling inhibitors [Hayashi et al., 201110]. […] Further manipulation of the micropattern differentiation conditions may give novel insights into the unique requirements of these different pluripotent cell states (Figure 1) for organized cell fate specification.”

"One of the conclusions is that EpiSCs are 'functionally restricted in developmental potential' (Results). However, EpiSCs can differentiate to a lot of cell types (see Huang et al., 2012): is there an argument that this micropattern culture has not fully revealed the spectrum of EpiSC differentiation?"

We thank the reviewers for this comment. We agree that chimaera data has shown that EpiSCs are not inherently restricted in their developmental potential. We now removed that statement and clearly reference the chimaera data within the Discussion (ninth paragraph).

As discussed above, further protocol modifications might permit EpiSC differentiation towards a wider spectrum of cell fates as observed in vivo (see above comments). For example, as EpiSCs can be maintained in a variety of culture conditions [1] and, individual EpiSC lines display distinct functional biases [2], different EpiSC lines, EpiSC culture conditions or micropattern differentiation conditions could result in the specification of distinct cell fates. We believe that we have established the micropattern assay as a quantitative read out of cellular state, complementary to single cell or population assays since it focuses on cell communication and pattern formation, that could be used to address these questions. While it would be beneficial to investigate the effect of these variables on cell fate specification, it is beyond the current scope of this paper.

Furthermore, while EpiSCs did not exhibit obvious spatial patterning of cell fates when differentiated in isolation, it may be possible that they can efficiently pattern cell fates when mixed with other cell types on the micropatterns, such as EpiLCs, in a situation that more closely resembles in vivo chimaera assays.

“More specifically, the formation of anterior neural fates was not extensively explored in this analysis. The conclusion of this study that EpiLCs are biased towards posterior embryonic and extra-embryonic mesodermal fates suggests that EpiSCs may be biased towards more anterior fates. This raises the possibility that EpiSCs may express functions normally found in the anterior extra-embryonic visceral endoderm. This possibility was not definitively excluded by the analysis presented here that is based on a restricted number of antibodies and did not employ genome wide analytical tools limiting the number of cellular states that might be detected. "

The conclusion of the study is not that EpiLCs are biased towards posterior cell fates but that the conditions that they are exposed to (+BMP or –BMP) promote distinct cell fates. EpiLCs can form posterior cell fates in the presence of BMP and anterior PS cell fates in the absence of BMP. Conversely, the EpiSCs in this study form anterior cell fates in the presence or absence of BMP.

As EpiLCs and EpiSCs are derived from or represent the post-implantation embryonic epiblast, which does not give rise to extraembryonic trophectoderm or endoderm, it is unlikely that the micropattern system contains extraembryonic endoderm or trophectoderm cell types. We have added this to the text in order to clarify the reasoning:

“For the mouse micropattern differentiation we utilized pluripotent EpiLCs, corresponding to the Epi cells of the embryo, and thus the system likely lacked the neighboring extraembryonic cell types and the signals that they provide.”

While we have carried out a thorough coexpression analysis of a panel of markers within individual cell populations present in the micropatterns and catalogued the cell populations that are present, the reviewers are correct that it is not possible to definitively assign an in vivo correlate to all cell populations present within the micropatterns. For example, it is problematic to distinguish between anterior primitive streak and axial mesoderm states that have a similar transcriptional signature. We will develop tools and protocols to facilitate the sorting of the cell populations identified in this current study for global transcriptional analysis and mapping to available embryonic data in future studies. We have now added a sentence to this affect:

“FOXA2/BRACHYURY-coexpressing cells observed in the absence of BMP (Figure 7) may correspond to a subpopulation of anterior PS cells (Figure 3) or alternatively AxM cells that have downregulated Gsc and Hhex. Global transcriptional analysis may be required to resolve these possibilities.”

Reviewer 2 requests some clarification regarding the unique outcomes of micropatterned cultures described here compared to other previous results."Results: 'Micropatterned cultures do not contain extraembryonic cell types (namely primitive endoderm or trophectoderm derivatives), they instead formed radially symmetric cell fate'. However, other 3d culture systems (e.g. van den Brink et al., 2014) generate asymmetry without extraembryonic cell types. Can the authors comment on how these differences could arise?"

We thank the reviewer for this comment. In vivo, the asymmetry of marker expression within the epiblast is initiated by signals from the extraembryonic trophectoderm and endoderm. It has previously been shown that, in the absence of these asymmetric extraembryonic signals, the epiblast no longer initiates asymmetric marker expression [3-6]. Although in vivo the source of asymmetry is the extraembryonic lineages, this does not mean that this is the only possible source of asymmetry in other contexts, for example in in vitro assays.

The reviewers draw attention to other studies where asymmetric marker expression has been demonstrated in vitro in the absence of extraembryonic cell types. However, those studies have fundamental differences in their methodology compared to the micropattern system. Pluripotent stem cell colonies differentiated on the disc-shaped micropatterns are confined in size and shape and begin as an epithelial monolayer exposed to a homogeneous medium. As such, there is no morphological or presumably signaling asymmetry. In contrast, the study by van den Brink et al. [7], referenced by the reviewers, involves three-dimensional embryoid body-like aggregates, assembled without size or morphological constraints and without constraints on growth/expansion during further differentiation. It is therefore likely that the aggregates are not morphologically symmetrical and, as a result, that individual cells may be exposed to distinct signaling environments and consequently initiate asymmetric signaling and marker expression.

In support of this hypothesis, when EpiLCs were differentiated on smaller diameter micropatterns, the width to height ratio of the colony was affected so that three-dimensional embryoid-body-like structures were formed. During the differentiation, these structures show morphological asymmetry that also correlated with a loss of marker radial symmetry. We have now added these results to the manuscript (Figure 5) and discuss this within the text:

“It should be noted that, on smaller diameter micropatterns, the width to height ratio of colonies was altered such that 80-140 μm diameter colonies generated taller, embryoid body-like aggregates. Over time, these three-dimensional structures exhibited morphological asymmetries (Figure 5), which may explain the loss of radial symmetry in marker expression.”

We have also now reworded the Discussion to further clarify this point:

“In vivo, localized signaling from the extraembryonic tissues, notably the anterior visceral endoderm (AVE), induces molecular asymmetries within the bilaterally symmetrical Epi leading to anterior-posterior axis establishment [Stower and Srinivas, 2014]. […] However, as these structures are not geometrically uniform, the polarized marker expression likely stems from initial morphological asymmetries and there is still no evidence of spontaneous symmetry breaking within geometrically uniform structures.”

All reviewers note some concerns about the identification of specific cell types. Reviewers 1 and 2 suggest that citation of current literature might help support the identification of particular cell types. Reviewer 3 raises a number of specific queries concerning particular markers; like reviewer 2 this individual is concerned to know whether particular markers were assessed at specific stages. It might be helpful to present the information on the markers in a somewhat more organized form. Figure S8 could be revised (for some reason I could not find the legend to this figure) to indicate which markers were tested at what stage, to indicate what figures show expression in the micropatterned cultures as well as the embryos, to include callouts to those data which are not identified with a particular figure in the manuscript, to show what the levels in the heatmap mean, and to include references from the literature where applicable. Excel might be the appropriate format for presenting this information."Since there was no attempt to characterise the cultured cells further, either through molecular techniques (e.g. single cell gene expression analysis using QRT-PCR or RNA-seq) or through studies of their further differentiation into appropriate lineages, the validation of these markers is particularly important. It is not easy to assess the fidelity with which the in vitro system replicates cells in the embryo on this basis of a few markers; it is possible that global examination of gene expression patterns might not support the identification of cell types, or that the cells in vitro might not have the developmental potential implied by the labels the authors assign to them. These limitations are shared by almost all studies of in vitro patterning to date, and at some stage, the field will have to progress beyond this level. At least, the basis for claiming that 2 or 3 markers can uniquely identify the cell types of interest (as tabulated in Supplemental Figure 3 and Figure 8) should be clearly specified: is it the immunostaining in the manuscript, or can the unique identifiers be further supported by gene expression data from the literature. Supplementary Figure 8 is quite critical, linking the data to assignment of cellular identities. Perhaps the supplemental material could group together the embryo images with staining identifying particular cell types through various stages. I found it awkward to keep flipping back and forth to check this information."

To address the reviewers’ concerns, we have supplied Supplementary file 1 and Supplementary file 2 comprising all embryo data used in this study including wholemount confocal images, confocal optical sections and cryosections of the markers utilized. Where we had used reporter ESC lines within the micropatterns and did not have access to good antibodies (for example for *Hhex, Goosecoid* and *Sox1*) or not all gastrulation stages were analyzed for a particular marker, we have supplied literature references indicating published expression data for those stages.

Furthermore, we have replaced the previous supplementary Figure 8 with an Excel file as suggested. All embryo data is summarized in Supplementary file 1 indicating expression (1), low or heterogeneous expression (0.5) or no expression (0). Supplied figure references indicate where the data can be found in the manuscript, additional data file or literature. We hope that this comprehensive and logical organization of the data now provides all of the necessary information in an easily searchable format.

See also these specific comments on markers for cell identification:Reviewer 1:1) Subsection “EpiLCs recapitulate differentiation and spatial organization of posterior cell fates”. Figure 2 what are the Sox2 positive cells in the extraembryonic region?

The strong *SOX2* signal present within the extraembryonic region, as in Figure 2, represents non-nuclear staining within the extraembryonic visceral endoderm. This has been noted in the figure legend for Figure 2. Additionally, higher resolution cryosection images have now been provided in the Supplementary file 2 and clear non-nuclear signal highlighted with a blue arrowhead. Similar non-nuclear background is observed in the extraembryonic visceral endoderm with other antibodies, including CDX2, OCT6 and BRACHYURY, and may represent an artifact related to the vacuolated structure of these cells [8]. However, as we are not analyzing extraembryonic endoderm or trophectoderm cell types, this does not affect cell fate assignments in this study.

2) Figure 2. What staining is shown in the top two panels of Figure 2?

The top 2 panels of Figure 2 show a merge of the markers indicated below. As this may have been unclear, we have now added a sentence to this effect into the figure legend: “Upper 2 panels represent a merge of the markers shown below.”

3) Subsection “EpiLCs recapitulate differentiation and spatial organization of posterior cell fates”. Figure 2 – what are the Cdx2 positive cells and what is the basis for their identification?

CDX2 positive cells most likely represent extraembryonic mesoderm based on a process of elimination of other markers and cell types. This reasoning is in detail in the Results section (subsection “FGF/ACTIVIN/WNT/BMP triggers spatially organized posterior fate specification”, fifth and sixth paragraphs).

4) Subsection “EpiLCs recapitulate differentiation and spatial organization of posterior cell fates”, fifth paragraph. Figure 3—figure supplement 1 – not clear that Otx2 is expressed in posterior epiblast from this image; indicate where is posterior.

OTX2 staining is reduced in the posterior Epi relative to the anterior Epi. However, we still observed a nuclear signal. Cryosections have now been supplied within Supplementary file 2 to further clarify this. We have now also provided a number of references [9-11] that show evidence (RNA in situ, immunostaining and transcriptional analysis) for low OTX2 expression within the posterior epiblast including the recent spatial transcriptome analysis of E7.0 embryos from Peng et al. [9] and clear immunostaining showing cells that coexpress OTX2 and FOXA2 within the mid-distal posterior Epi [10]:

“In agreement with a recent spatial transcriptional analysis of gastrulating mouse embryos [Peng et al., 2016], our immunostaining data suggested that posterior Epi cells may also express low levels of OTX2 (Figure 3—figure supplement 1). At E7.0-E7.5, a fraction of distal posterior Epi cells begin to express FOXA2 (Figure 3) [Burtscher and Lickert, 2009], some of which also express OTX2 [Peng et al., 2016; Engert et al., 2013].”

5) Figure 3 – indicate in the legend what the little red dots do i.e. delineate region of interest.

The dashed lines indicate the region of interest and this has now been added to the figure legend for Figure 3 as well as for Figure 2.

6) High resolution high magnification images of the embryos should perhaps be compiled somewhere to support the use of the markers in question.

We have now supplied all embryo data in Supplementary file 1 and Supplementary file 2 This contains cryosections that display marker expression at higher magnification.

Reviewer 2:Along similar lines it would make the study more influential if the contemporary literature in early mouse development was more precisely cited. For example, the differences in the expression patterns of Nanog and Brachyury in the gastrulating mouse embryo that are central to the current report have been previously reported raising the important possibility that Nanog may have distinct functions in the inner cell mass and the primitive streak.

References for the expression pattern of each marker utilized in this study have now been added throughout the text. We have also included references to recent Geo-Seq data [9] where expression within certain domains has not been extensively characterized in the literature e.g. Otx2 expression within the posterior epiblast.

Reviewer 3:1) It would be good to re-check all the specifics in the rubric. Some queries that arose from using this as the overview to all the data, including:a) Was Nanog immunostaining done at all stages? It is listed as posterior epiblast throughout but I don't see the immunostains this refers to.

NANOG staining was done at E6.5-6.75, E7.0-7.5 and E7.75-8.0. We have now added all of this data including cryosections to Supplementary file 1 and Supplementary file 2 and the references to the location in the main figures and can be found within Supplementary file 1.

b) Sox1-GFP should be added to the rubric to identify anterior epiblast.

This has now been added to Supplementary file 1 with a reference to the expression pattern as described in the literature.

c) The rubric lists 'Figure 5' as a source of information – I couldn't find panel 5I.d) Definitive endoderm at E7.5-8.0 should list Figure 3—figure supplement 1 as a source.

We thank the reviewer for bringing these errors to our attention and have now updated all information provided within Supplementary file 1.

e) Was OTX2 stain done in E7.0-7.25 embryos?

OTX2 immunostaining was carried out in E6.25, E7.5 ant E7.75 embryos but not E7.0-7.25 embryos. All data is shown in Supplementary file 1 and literature references supplied for the missing E7.0-7.25 time point within the accompanying Excel file.

2) Some of the conclusions may need qualifying. For example check 'cells that coexpressed GATA6 and OTX2 (Figure 3—figure supplement 1), corresponding to Mesoderm 1': GATA6 and OTX2 are also expressed in endoderm. Is the argument that this is not endoderm because the endoderm only later expresses OTX2? Timings of individual markers may be a bit different in vitro. However, focusing on the most likely cell type is correct; just try and avoid overstating the certainty of a conclusion.

To address this issue, we have reworded the text throughout to indicate that these are the most likely, rather than definitive, cell fate assignments based on marker combinations.

As the reviewer states, GATA6 and OTX2 are coexpressed in both mesoderm and in definitive endoderm (DE) cells. However, the GATA6-expression domain is devoid of the DE marker FOXA2 (found only in the center of micropatterns in the presence of BMP, Figure 3). We also added a new figure panel to show that, furthermore, SOX17 (also a DE marker as well as being expressed in the extraembryonic mesoderm) is expressed in an almost entirely mutually exclusive fashion with GATA6 (Figure 3—figure supplement 2). SOX17 and FOXA2 are also expressed within distinct colony domains (outer and central domains respectively, Figure 3). Hence, while GATA6, OTX2, FOXA2 and SOX17 are all DE markers and are all expressed within the micropatterns, other than GATA6 and OTX2, their expression does not overlap. Based on these observations we had assigned GATA6/OTX2 cells as mesoderm rather than DE. We have modified the text to explain this reasoning in more detail:

“During the micropattern differentiation, multiple DE-associated markers were expressed, namely FOXA2, SOX17, GATA6 and OTX2. In vivo these markers are coexpressed within DE cells (Figure 3—figure supplement 1) while in the micropattern differentiation FOXA2, SOX17 and GATA6 were expressed in a mostly mutually exclusive manner (Figure 3—figure supplement 2), hence they marked separate non-DE populations.”

We have also now added additional data to strengthen this assignment. We utilized a double reporter cell line for *Goosecoid* and *Hhex* [12], which are markers coexpressed within DE and axial mesoderm. In the presence of BMP, *Goosecoid* was not expressed. This confirms our initial observations, suggesting an absence of DE and axial mesoderm anterior cell fates. In the presence of BMP, we did observe *Hhex* expression although *Hhex* also marks haematopoetic cells arising in the allantois. This data has now been added within Figure 3—figure supplement 2 (see subsection “FGF/ACTIVIN/WNT/BMP triggers spatially organized posterior fate specification” seventh paragraph).

Alternatively, when we remove BMP, we observed coexpression of *Goosecoid* and *Hhex* indicating the presence of DE or axial mesoderm or both. We have also added this data to the manuscript (Figure 7—figure supplement 1) (see subsection “The absence of BMP allows DE and AxM specification”, second paragraph).

References:

1) Morgani, S., J. Nichols, and A.K. Hadjantonakis, The many faces of Pluripotency: in vitro adaptations of a continuum of in vivo states. BMC Dev Biol, 2017. 17(1): p. 7.

2) Bernemann, C., et al., Distinct Developmental Ground States of Epiblast Stem Cell Lines Determine Different Pluripotency Features. Stem Cells, 2011. 29(10): p. 1496-1503.

3) Nowotschin, S., et al., The T-box transcription factor Eomesodermin is essential for AVE induction in the mouse embryo. Genes & Development, 2013. 27(9): p. 997-1002.

4) Migeotte, I., et al., Rac1-Dependent Collective Cell Migration Is Required for Specification of the Anterior-Posterior Body Axis of the Mouse. Plos Biology, 2010. 8(8).

5) Kimura-Yoshida, C., et al., Canonical Wnt signaling and its antagonist regulate anterior-posterior axis polarization by guiding cell migration in mouse visceral endoderm. Developmental Cell, 2005. 9(5): p. 639-650.

6) Ding, J.X., et al., Cripto is required for correct orientation of the anterior-posterior axis in the mouse embryo. Nature, 1998. 395(6703): p. 702-707.

7) van den Brink, S.C., et al., Symmetry breaking, germ layer specification and axial organisation in aggregates of mouse embryonic stem cells. Development, 2014. 141(22): p. 4231-42.

8) Wada, Y., G.H. Sun-Wada, and N. Kawamura, Microautophagy in the visceral endoderm is essential for mouse early development. Autophagy, 2013. 9(2): p. 252-254.

9) Peng, G.D., et al., Spatial Transcriptome for the Molecular Annotation of Lineage Fates and Cell Identity in Mid-gastrula Mouse Embryo. Developmental Cell, 2016. 36(6): p. 681-697.

10) Engert, S., et al., Wnt/β-catenin signalling regulates Sox17 expression and is essential for organizer and endoderm formation in the mouse. Development, 2013. 140(15): p. 3128-3138.

11) Auman, H.J., et al., Transcription factor AP-2 γ is essential in the extra-embryonic lineages for early postimplantation development. Development, 2002. 129(11): p. 2733-2747.

12) Villegas, S.N., et al., PI3K/Akt1 signalling specifies foregut precursors by generating regionalized extra-cellular matrix. *ELife*, 2013. 2: p. e00806.

13) Warmflash, A., et al., A method to recapitulate early embryonic spatial patterning in human embryonic stem cells. Nat Methods, 2014. 11(8): p. 847-54.

14) Winnier, G., et al., Bone Morphogenetic Protein-4 Is Required for Mesoderm Formation and Patterning in the Mouse. Genes & Development, 1995. 9(17): p. 2105-2116.

15) Mishina, Y., et al., Bmpr encodes a type I bone morphogenetic protein receptor that is essential for gastrulation during mouse embryogenesis. Genes Dev, 1995. 9(24): p. 3027-37.

16) Beppu, H., et al., BMP type II receptor is required for gastrulation and early development of mouse embryos. Developmental Biology, 2000. 221(1): p. 249-258.